

# Characteristics of the Derived Energy Dissipation Rate using the 1-Hz Commercial Aircraft Quick Access Recorder (QAR) Data

Soo-Hyun Kim[1], Jeonghoe Kim[1], Jung-Hoon Kim[1], and Hye-Yeong Chun[2]

[1]School of Earth and Environmental Sciences, Seoul National University, Seoul, South Korea
[2]Department of Atmospheric Sciences, Yonsei University, Seoul, South Korea

*Correspondence to*: Prof. Jung-Hoon Kim (jhkim99@snu.ac.kr)

**Abstract.** The cube root of the energy dissipation rate (EDR), as a standard reporting metric of atmospheric turbulence, is estimated using 1-Hz quick access recorder data from Korean-based national air carriers with two different types of aircraft [Boeing 737 (B737) and B777], archived for 12 months from January to December 2012. Various EDRs are estimated using

zonal, meridional, and derived vertical wind components, and the derived equivalent vertical gust (DEVG). Wind-based EDRs are estimated by (i) second-order structure function (EDR1), (ii) power spectral density (PSD), considering the Kolmogorov's -5/3 dependence (EDR2), and (iii) maximum-likelihood estimation using the von Kármán spectral model (EDR3). DEVG-based EDRs are obtained mainly by vertical acceleration with different conversions to EDR using (iv) the lognormal mapping technique (EDR4) and (v) the predefined parabolic relationship between the observed EDR and DEVG (EDR5). For the EDR1,

second-order structure functions are computed for zonal, meridional, and vertical wind within the defined inertial subrange. For the EDR2 and EDR3, individual PSDs for each wind component are computed using the Fast Fourier Transform over every 2-minute time window. Then, two EDR estimates are computed separately by employing the Kolmogorov-scale slope (EDR2) or prescribed von Kármán wind model (EDR3) within the inertial subrange. The resultant EDR estimates from five different methods follow a lognormal distribution reasonably well, which satisfies the fundamental characteristics of

atmospheric turbulence. Statistics (mean and standard deviation) of log-scale EDRs are somewhat different from those found in a previous study using a higher frequency (10 Hz) of in situ aircraft data in the United States, likely due to different sampling rates, aircraft types, and locations. Finally, five EDR estimates capture well the intensity and location of three strong turbulence cases that are relevant to clear-air turbulence (CAT), mountain wave turbulence (MWT), and convectively induced turbulence (CIT), with different characteristics of the observed EDRs: 1) zonal (vertical) wind-based EDRs are stronger in the CAT (CIT)

case, while MWT has a peak of EDRs in both zonal and vertical wind-based EDRs, and 2) the CAT and MWT cases occurred by large-scale (synoptic-scale) forcing have more variations in EDRs before and after the incident, while the CIT case triggered by smaller mesoscale convective cell has an isolated peak of EDR.




## 1 Introduction

Turbulence encounters are major threats to the aviation industry that can result in serious structural damage to aircraft and injuries to passengers and flight crew (Sharman and Lane, 2016; Gultepe et al., 2019). To mitigate the risk associated with turbulence, the use of a large number of reliable turbulence observations is essential, both for extending our understanding of turbulence and accurately forecasting it. Routinely, turbulence observations are provided in the form of verbal reports by pilots (PIREPs). In PIREPs, information is given on turbulence intensity (null, light, moderate, severe, extreme), time, and location (longitude, latitude, and flight levels) for turbulence encounters. However, the turbulence intensity in PIREPs is determined by a pilot's subjective assessment of the aircraft response to turbulence encounters, and this may introduce uncertainty into turbulence information (Schwartz, 1996; Sharman et al., 2014). Considering that null reports are not routine, PIREPs are not sufficient for constructing reliable maps of turbulence globally. To address these issues, objective aircraft-based turbulence observations have been widely used in the research community via collaborations with airline industries (e.g., Haverdings and Chan, 2010; Kim and Chun, 2012, 2016; Sharman et al., 2012, 2014; Gill, 2014; Kim et al., 2017, 2018, 2020, 2021a; Sharman and Pearson, 2017) and field experiments using research aircraft (e.g., Koch et al., 2005; Strauss et al., 2015; Williams and Meymaris, 2016; Bramberger et al., 2018).

There are two representative turbulence metrics based on in situ aircraft measurements: the derived equivalent vertical gust velocity (DEVG) (Hoblit, 1998), and the cube root of the energy dissipation rate (EDR) (Sharman et al., 2014). These two metrics are included in some of the Aircraft Meteorological Data Relay (AMDAR) data as turbulence information (WMO, 2003). Given that the DEVG is a gust-load transfer factor and is not a direct turbulence estimate, the EDR is more useful and preferred over the DEVG for turbulence forecasting applications and turbulence detection (Sharman et al., 2014; Kim et al., 2020). Indeed, the EDR is designated as a standard reporting metric of turbulence by the International Civil Aviation Organization (ICAO) (ICAO, 2001, 2010). The National Center for Atmospheric Research (NCAR) developed the EDR algorithm based on aircraft vertical acceleration or derived vertical velocity (Sharman et al., 2014; Cornman, 2016), and it has been implemented on several international air carriers, such as United Airlines, Delta Air Lines, and South West Airlines. This has made it possible to produce automatic EDR measurements. Haverdings and Chan (2010) also developed an off-line vertical velocity-based EDR algorithm and tested it on in situ data from Hong Kong-based airline fleets.

The in situ EDR reports of turbulence have been widely used in many case studies on turbulence detection (e.g., Kim and Chun, 2012; Trier et al., 2012; Trier and Sharman, 2018; Zovko-Rajak et al., 2019) and in performance evaluations of turbulence forecasts (e.g., Kim et al., 2015, 2018, 2019a; Pearson and Sharman, 2017; Sharman and Pearson, 2017). However, these in situ EDR reports are only available by negotiation with commercial airlines, which limits the volume and extent of turbulence observations. In addition, considering that the minimum recommended sampling frequency for flight parameters (e.g., angle of attack, pitch, and roll) is 4 Hz (Sharman et al., 2014), investigations based on EDR estimates have been conducted under restrictions, with enough data to satisfy the minimum requirements. Although there have been attempts to estimate the EDR using other sources of data, such as weather radar, lidar, radiosonde, and sonic anemometers (e.g., Muñoz-





Esparza et al., 2018; Bodini et al., 2019; Ko et al., 2019; Kim et al., 2021a), measurements by commercial civil aviation aircraft worldwide remain the most important source of information from which to estimate turbulence intensity and climatology at cruising altitudes. To complement the limited availability of global turbulence observations, Kim et al. (2020) retrieved the EDR from the DEVG, originating from AMDAR data, using two conversion methods–one based on the lognormal mapping

technique of Sharman and Pearson (2017), and the other based on the predefined parabolic relationship between the EDR and the DEVG constructed by Kim et al. (2017). Given that the DEVG is based on the gust-load transfer factor, additional approaches to retrieve direct turbulence estimates using aircraft measurements should be considered. It is also noted that Kopeć et al. (2016) proposed methods to estimate the EDR using the 1-Hz sampling rate of navigational information of commercial aircraft in the form of Mode-S Enhanced Surveillance (EHS) and Automatic Dependent Surveillance-Broadcast (ADS-B), to

complement the measurements obtained from on-board devices. As an additional data source of turbulence, turbulence estimates using the Mode-S EHS and ADS-B can be valuable additions, however, it remains important to maximize the utilization of in situ data from aircraft measurements.

For additional use of commercial aircraft on-board data, in the current study, we used the 1-Hz commercial aircraft Quick Access Recorder (QAR) data for calculating the EDR, based on five different EDR estimation methods. This included

the EDR conversion from the DEVG proposed by Kim et al. (2020). With respect to both data link and storage, the use of 1-Hz flight data to estimate the EDR is cost effective compared to in situ EDR measurements with a higher sampling rate (e.g., 8 Hz or 10 Hz). Despite the advantage to the 1-Hz (coarser frequency) data, higher frequency (e.g., 8 Hz or 10 Hz) data is still required to represent highly transient and variant characteristics and intensity of turbulence more accurately. In this manner, this higher frequency NCAR EDR algorithm has been developed and implemented in some of United States (US)-based

commercial airline aircraft, and will be extended to more airliners worldwide in future (e.g., Sharman et al., 2014; Cornman, 2016). The main purpose of this study is to examine the feasibility of various objective EDR estimations using the 1-Hz (coarser frequency) flight data for possible sources of atmospheric turbulence in cruising altitudes. This will be eventually useful for applying these EDR estimation methods to the 1-Hz real-time data in future. Possible candidates for the real-time data are the navigational information of commercial aircraft such as the Mode-S EHS and ADS-B. The QAR data used in the

current study were not the real time data but the retrieved post-flight data obtained from Korean Air Lines (KAL) Boeing (B) 737 and B777 aircraft recorders over a 12-month period (January–December 2012). Note that Kim and Chun (2016) already used the same dataset and conducted quality control to calculate the DEVG. The feasibility of using EDR estimates calculated from the QAR data is evaluated using selected moderate-or-greater (MOG)-level turbulence cases. This paper is organized as follows. A description of the QAR data and estimation of wind velocity are provided in Sect. 2. In Sect. 3, the descriptions of

EDR estimation and EDR statistics are provided. In Sect. 4, the EDR estimates are examined with selected MOG-level turbulence cases. In Sect. 5, a summary and discussion are provided.





## 2 Data and methodology

### 2.1 QAR data

The QAR data used in the current study were obtained from the on-board recorders of 99 flights (B737: 64 flights and B777: 35 flights) operated by KAL from January to December 2012. The aircraft flight parameters used in the current

study were recorded every second (1 Hz) for both the B737 and B777. Because the 1-Hz wind direction and wind speed of the B737 records had the same values within a 4-second time window, the wind direction and wind speed of the B737 records seem to have 1/4-Hz sampling frequency. For the EDR estimations, we use two groups of EDRs. The first is wind-based EDRs using the three wind components (zonal, meridional, and vertical wind components). The second is the DEVG-based EDRs calculated using the time series of aircraft vertical acceleration, aircraft mass, airspeed, altitude in flight levels, and aircraft

type (Truscott, 2000; Kim and Chun, 2016; Kim et al., 2020). The details of the DEVG calculation can be found in Kim and Chun (2016).

Kim and Chun (2016) conducted quality control procedures to remove erroneous data related to the aircraft vertical acceleration, static air temperature, altitude, and aircraft mass. Additionally, the lower limit of altitude was set to 15 kft in order to remove a misleading value obtained while aircraft are manoeuvring. Thus, the current study uses the QAR data above

15 15 kft qualified by Kim and Chun (2016), and examines the B737 and B777 QAR data separately. The total number of B737 and B777 QAR data above 15 kft useful for analysis is 264,867 and 1,065,855, respectively.

Figure 1 shows the horizontal distribution of the number of B737 and B777 aircraft data collected over 12 months (from January to December 2012) above 15 kft, accumulated within a 0.5°×0.5° horizontal grid box. As shown in Fig. 1, the B737 (mid-size aircraft) data used in the current study had a time series of flight parameters over some of Asia (relatively

shorter flight routes), while the B777 data included relatively longer flights over some of Asia, Oceania, Europe, North America, and South America. For both aircraft types, the number of QAR data collected along flight routes over some of the Pacific Ocean, North America, and South America is much smaller than that over some of Asia, such as Hong Kong, Taiwan, and South Korea. Given that statistical analyses on the aircraft-based EDR estimates over Asia have not been sufficient, the current study can provide valuable information on the characteristics of turbulence over Asia, together with suggesting the

feasibility of EDR estimation methods using 1-Hz sampling of aircraft measurements. In the following section, wind estimation for deriving the wind-based EDR estimation is described.

### 2.2 Wind estimation

Using wind direction and wind speed of the on-board aircraft data, the zonal (U) and meridional (V) wind is computed. The vertical wind (W) is estimated as the difference between the true airspeed and the aircraft inertial vertical velocity (IVV)

(e.g., Lenschow, 1972; Sharman et al., 2014):

$$W = V_T ( \sin \alpha_b \cos \theta \cos \varphi + \sin \beta \cos \theta \sin \varphi - \cos \alpha_b \cos \beta \sin \theta ) - IVV - M \dot{\theta} \cos \theta, \tag{1}$$



where $V_T$ is the true airspeed in m s$^{-1}$, $\alpha_b$, $\theta$, $\varphi$, and $\beta$ are the body-axis angle of attack, the pitch angle, the roll angle, and the sideslip angle, respectively, $\dot{\theta}$ is the pitch rate, and $M$ is the distance between the measurement location of the angle of attack and the aircraft center of gravity.

Given that the sideslip ($\beta$) can be negligible (Haverdings and Chan, 2010) and the rightmost term related to the pitch

rate ($\dot{\theta}$) contributes less to the resultant EDR estimates (Sharman et al., 2014), Eq. (1) can be approximated to a simpler form (Haverdings and Chan, 2010; Sharman et al., 2014):

$$W = V_T \left( \sin \alpha_b \cos \theta \cos \varphi - \cos \alpha_b \sin \theta \right) - IVV. \tag{2}$$

The $\alpha_b$ is computed from the measured right and left vane angle of attack ($\alpha_R$ and $\alpha_L$, respectively) as

$\alpha_b = a_0 + a_1 \, \bar{\alpha}$ and

$\bar{\alpha} = (\alpha_R + \alpha_L)/2, \tag{3}$

where $\bar{\alpha}$ is the locally observed angle of attack that averages the $\alpha_R$ and $\alpha_L$, and $a_0$ and $a_1$ are calibration coefficients. Since the pitch angle $\theta$ can be assumed to equal the body-axis angle of attack $\alpha_b$ during steady flight (Williams and Marcotte, 2000; Drüe and Heinemann, 2013; Sharman et al., 2014), Eq. (3) can be written as

$\theta = \alpha_b = a_0 + a_1 \, \bar{\alpha}. \tag{4}$

In Eq. (4), the calibration coefficients $a_0$ and $a_1$ are computed through the least-squares linear regression between $\theta$ and $\bar{\alpha}$ following Sharman et al. (2014). The slope and y-axis intercept yield from the linear regression of $\bar{\alpha}$ and $\theta$ are assigned to calibration coefficients $a_1$ and $a_0$, respectively. A similar approach was conducted by Williams and Marcotte (2000) and Drüe and Heinemann (2013).

Figure 2 shows scatter density plots of the measured angle of attack ($\bar{\alpha}$) and aircraft pitch angle ($\theta$), and linear

regression fits for the B737 and B777. To objectively retrieve the intensity of atmospheric turbulence using the aircraft data (especially using derived W), the best relationship between the angle of attack ($\bar{\alpha}$) and aircraft pitch angle ($\theta$) for estimating the derived W should be found. Because two parameters (pitch angle and angle of attack) are highly sensitive to the navigation of aircraft, any data where the altitude rate is less than or equal to 10 ft/s and the altitude is greater than or equal to 15 kft are used exclusively. Using this criterion, we found that most of the flight data [81% of the B737 data (214,450 reports) and 94%

of the B777 data (1,004,037 reports)] are in the cruising mode of the steady flights, which are eventually used to construct the best linear regression between the angle of attack ($\bar{\alpha}$) and aircraft pitch angle ($\theta$). As a result, both B737 and B777 QAR data yield representative linear fits ($a_0 = 3.154$ and $a_1 = 0.594$ for B737 and $a_0 = 2.096$ and $a_1 = 0.517$ for B777). These calibration coefficients are used to calculate the vertical wind (W) using Eq. (2). This derived vertical wind is used to compute the EDR, together with zonal and meridional winds and the DEVG.





## 3 EDR estimations

The EDR is estimated from 1-Hz B737 and B777 QAR data, as shown in Fig. 1 (264,867 and 1,065,855 reports from B737 and B777 recorders, respectively) using five methods: three methods calculate the EDRs from the wind components based on the inertial dissipation method (IDM, also known as inertial range technique) (Champagne et al., 1977), and two

methods estimate the EDR from the DEVG, based on the lognormal mapping scheme of Sharman and Pearson (2017) and the prescribed best-fit curve of Kim et al. (2017, 2020). Kim et al. (2020) showed that statistical occurrence of turbulence by DEVG-derived EDRs calculated from AMDAR data is similar to that from in situ EDR measurements. Descriptions of the five methods are provided in the following sections.

### 3.1 EDR estimation using the IDM (EDR1, EDR2, and EDR3)

In the EDR estimation based on the IDM, Taylor's frozen turbulence hypothesis (Hinze, 1975) is invoked to express the EDR in a temporal domain. It is noted that the high true airspeed of aircraft relative to the perturbation of velocity ($u'=u-\overline{V_T}$, where overbar is mean true airspeed) can hold this hypothesis to be valid ($u'/\overline{V_T} \ll 1$). The EDR can be estimated from the three methods utilizing (i) the second-order structure function (e.g., Frehlich and Sharman, 2004), (ii) the power spectral density (PSD), and (iii) the maximum-likelihood estimation using the von Kármán spectral model (Sharman et al., 2014).

First, applying Kolmogorov's second hypothesis of similarity (Kolmogorov, 1941) and Taylor's hypothesis, the EDR can be expressed in terms of temporal increment ($\tau$) of wind velocity, as

$$EDR = \left(\frac{1}{\overline{V_T}}\right)^{1/3} \left[\frac{\overline{D_i(\tau)}}{\tau^{-2/3}C_K}\right]^{1/2}, \text{ and}$$

$$D_i(\tau) = \langle [u_i(t+\tau)-u_i(t)]^2 \rangle. \tag{5}$$

Here, $\overline{V_T}$ is the averaged true airspeed for a 2-minute time window, $D_i$ is the second-order structure function of the

20 wind velocity component $u_i = [U, V, \text{ and } W]$ with a 2-minute window, $C_K$ is the Kolmogorov constant set as 0.52 for U and 0.707 for V and W (Wyngaard and Coté, 1971; Oncley et al., 1996; Strauss et al., 2015), and the overbar is the arithmetic average over the predefined inertial range ($2 \le \tau \le 5$ s) that is about 400–1,000 m of horizontal scale, considering airspeed is about 200 m s$^{-1}$. This is sufficiently small enough to feel atmospheric turbulent eddies that directly affect mid-size aircraft at cruising altitudes in the upper troposphere and lower stratosphere (UTLS) (e.g., Sharman et al., 2006, 2014).

Figure 3 shows an example of the computed second-order structure function ($D_i$) of meridional wind within 2 minutes (from 1128 UTC to 1130 UTC 11 October 2012) when strong turbulence is observed, with a change of the vertical acceleration of more than 0.8 g (g is the gravitational acceleration) and the DEVG of 5.555 m s$^{-1}$, which is categorized as moderate turbulence (Truscott, 2000). The structure function within the defined inertial range ($2 \le \tau \le 5$ s) closely follows the $\tau^{2/3}$ inertial range slope by the Kolmogorov turbulence hypothesis in the selected case. Although the degree of agreement between the

observed wind structure functions and theoretical slope may be different case by case and according to the wind component





(not shown), the observed structure functions calculated using 1-Hz QAR data represent, qualitatively, the property of turbulence within the inertial range. Finally, from this second-order structure function, the EDR can be calculated using Eq. (5), which is defined as EDR1. When the EDR is calculated from the zonal (meridional and derived vertical) wind velocity using this method, it is referred to as EDR1U (EDR1V and EDR1W, respectively).

Second, the EDR can be estimated by fitting the Kolmogorov slope of $k^{-5/3}$ to the observed power spectral density of the wind component $[S_i, (i = U, V, and W)]$ in the defined inertial range, and by assuming Taylor's frozen hypothesis:

$$EDR = \left(\frac{2\pi}{V_T}\right)^{1/3} \left[\overline{\frac{S_i(f)\,f^{5/3}}{C_K^{-1}}}\right],  \tag{6}$$

where a range of frequency $f$ ($f = \frac{V_T}{2\pi}k$, where $k$ is wavenumber) is between 0.2 and 0.5 s$^{-1}$, which corresponds to the inertial range defined in Eq. (5), and the overbar is the arithmetic mean over the data within the defined inertial range. The PSD of
each wind component is estimated by a fast Fourier transform with no window. Before computing the FFT, the wind data are tapered using a Welch window. The resultant EDR from Eq. (6) is referred to as EDR2 and the EDR2 from U, V, and W are labelled EDR2U, EDR2V, and EDR2W, respectively.

Third, the EDR is estimated using the maximum-likelihood estimation method (Sharman et al., 2014), which uses observed energy spectra ($S_{obs}$) and model spectra ($S_{model}$) given by

$$EDR = \left(\frac{1}{p_2 - p_1 + 1}\sum_{f=f_1}^{f_2}\frac{S_{obs}(f)}{S_{model}(f)}\right)^{1/2},  \tag{7}$$

where $f_1$ and $f_2$ are 0.2 and 0.5 s$^{-1}$, respectively and $p_1$ and $p_2$ are the lower and upper frequency indices, respectively. The von Kármán energy spectra (von Kármán, 1948; Mann, 1994) are used as the model spectra $S_{model}$. It is noted that the observed spectrum $S_{obs}$ and defined inertial range are the same as used in Eq. (6).

For zonal wind (U) data, the von Kármán wind model in the spatial domain is formulated as

$$F_{model}(k) = \frac{9}{55}\alpha\,\varepsilon^{2/3}\frac{1}{(L^{-2}+k^2)^{5/6}},  \tag{8}$$

where the empirical value of $\alpha$ is set to 1.6 (Sharman et al., 2014), L is the length scale, which is set to 669 m following Sharman et al. (2014), and $\varepsilon$ should be unity.

For meridional wind (V) and vertical wind (W) data, the von Kármán energy spectra is formulated as

$$F_{model}(k) = \frac{3}{110}\alpha\,\varepsilon^{2/3}\frac{3\,L^{-2} + 8\,k^2}{(L^{-2}+k^2)^{11/6}}.  \tag{9}$$

Taylor's turbulence hypothesis is applied to Eqs. (8) and (9) to convert the frequency ($f$)-domain-based spectrum ($S_{model}$) from the temporal ($k$)-domain spectrum. The resultant EDR from Eqs. (7) – (9) is referred to as EDR3, and the EDR3 derived using U, V, and W are described as EDR3U, EDR3V, and EDR3W, respectively.





Figure 4 shows an example of the PSD of zonal wind obtained from the QAR data at 1038 UTC 11 October 2012, when aircraft rarely experienced turbulence because of a change of the vertical acceleration of less than ~0.2 g. The von Kármán and Kolmogorov's theoretical $f^{-5/3}$ slopes for the energy spectrum, which are related to EDR2 and EDR3, respectively, are also included in Fig. 4. In general, the observed PSD follows well the theoretical -5/3 slopes. This result implies that the

1-Hz aircraft measured data can feasibly be used to estimate the EDR using PSD-based approaches. Further evaluation will be conducted through the case analysis in Sect. 4.

**3.2 EDR conversion using the prescribed best-fit function (EDR4)**

Kim et al. (2020) converted the EDR from the DEVG using the polynomial curve between the observed EDR and the DEVG constructed by Kim et al. (2017). Note that the observed EDR was computed using time series of Hong Kong-based

airlines data, based on the EDR algorithm of Haverdings and Chan (2010) (Kim et al., 2017). On a one-to-one basis, Kim et al. (2017) constructed the parabolic curve between the EDR and DEVG for each type of aircraft. For the EDR conversion, Kim et al. (2020) used the best-fit curve of Boeing aircraft (B747 and B777), which have a high correlation and accuracy, as

$$DEVG* = 0.0031\,(DEVG^2) + 0.0286\,(DEVG) + 0.0114, \tag{10}$$

where DEVG* is the remapped value to the EDR scale with units of $m^{2/3}\,s^{-1}$. The current study uses this best-fit curve following

Kim et al. (2020) to convert the EDR from the DEVG. It is noted that the same equation (Eq. 10) is applied to both 1-Hz B737 and B777 DEVG data. The resultant EDR is called EDR4.

**3.3 EDR conversion using the lognormal mapping technique (EDR5)**

Considering a lognormality of turbulence discussed in many previous studies (e.g., Nastrom and Gage, 1985; Frehlich, 1992; Cho et al., 2003; Frehlich and Sharman, 2004; Sharman et al., 2014; Kim et al., 2017, 2020), Sharman and Pearson

(2017) developed a lognormal mapping technique from numerical weather prediction (NWP)-based turbulence diagnostics with different physical meanings and units. This was designed such that each turbulence diagnostic climatologically corresponds to turbulence observations that follow a lognormal distribution (i.e., the random and chaotic nature of turbulence in the atmosphere). Assuming the lognormal behavior of turbulence, the simplest mapping between a raw diagnostic D and the EDR is applied as follows:

$$\ln(D*) = \ln(EDR) = a + b\ln(D),$$

$$a = \langle \ln(EDR) \rangle - b\langle \ln(D) \rangle = C_1 - b\langle \ln(D) \rangle,\ and$$

$$b = SD\ln(EDR)/SD\ln(D) = C_2/SD\ln(D), \tag{11}$$

where D* is the turbulence diagnostic remapped to the EDR scale, the angle bracket is an ensemble average, and $SD\ln(D)$ and $SD\ln(EDR)$ are a standard deviation (SD) of the natural logarithm of the turbulence diagnostic D and that of EDR





observations, respectively. Climatological values $C_1$ and $C_2$ are set to -2.953 and 0.602, respectively, as given in Sharman and Pearson (2017) and obtained by long-term in situ EDR estimates from US-based air carriers above a 20 kft flight level. Considering that the 1-year (2012) period of the QAR data used in this study overlaps the research period (from 2009 to 2014) of the dataset used in Sharman and Pearson (2017), it is considered that the use of climatological values of Sharman and

Pearson (2017) is acceptable. Although in recent days there are some efforts to update $C_1$ and $C_2$ for the low-level turbulence using high-frequency sonic anemometer mounted in the tall towers (e.g., Muñoz-Esparza et al., 2018; Kim et al., 2021b), to our knowledge at the present, there is no recent update on $C_1$ and $C_2$ for the upper level because it requires a large amount of high-frequency aircraft data for the EDR estimation.

As carried out by Kim et al. (2020), the turbulence diagnostic D in Eq. (11) is replaced with the DEVG estimates, as

$\ln(\text{DEVG*}) = a + b \ln(\text{DEVG})$,

$a = \langle \ln(\text{EDR}) \rangle - b \langle \ln(\text{DEVG}) \rangle = C_1 - b \langle \ln(\text{DEVG}) \rangle$, and

$b = \text{SD} \ln(\text{EDR}) / \text{SD} \ln(\text{DEVG}) = C_2 / \text{SD} \ln(\text{DEVG})$,                    (12)

Here, DEVG* is the remapped DEVG into the EDR scale. To obtain the mean and SD of $\ln(\text{DEVG})$, the lognormal fitting is conducted via the nonlinear least-squares fit, which uses the Levenberg-Marquardt algorithm (Moré, 1978). The data

used to calculate the PDF are binned with the samples that are greater than 10 reports in each bin among 50 bins.

Figure 5 shows the probability density functions (PDFs) of the DEVG in units of m s$^{-1}$, computed using both the B737 and B777 QAR datasets of Fig. 1 for the same period (12 months), and lognormal fits applied to the PDFs. The largest value of the DEVG satisfying the criteria of data binning is about 3.31 and 6.18 m s$^{-1}$ for B737 and B777, respectively (not shown). From the lognormal fits, the mean and SD of $\ln(\text{DEVG})$ are -2.323 and 1.031, respectively, for B737 and -2.768 and 1.180,

respectively, for B777, and these values are used to calculate the EDR. As the QAR data do not have a wide coverage, the seasonal and regional mean and SD are not considered. Although one-to-one comparison to Kim et al. (2020) is difficult due to the limitations in spatiotemporal coverage, the overall magnitudes of the mean and SD of the DEVG in the current study are smaller than in Kim et al. (2020). The resultant EDR calculated from Eq. (12) is called EDR5. More detailed results of comparison of PDFs with other EDR methods are given in the following sections.

**3.4 Intercomparison of the EDR estimates**

The current NWP-based turbulence forecasting methods (e.g., Sharman and Pearson, 2017; Pearson and Sharman, 2017; Kim et al., 2018, 2019a) use the lognormal mapping technique of Sharman and Pearson (2017) (Eq. 11 of the current study), which requires two important statistics (the mean and SD) of the observed EDR. For the mean and SD of the log-scale observed EDR [$\langle \ln(\text{EDR}) \rangle$ and $\text{SD} \ln(\text{EDR})$ of Eq. (11), respectively], Sharman and Pearson (2017) provided climatological

values [$C_1$ and $C_2$ of Eq. (11)] calculated using in situ equipped EDR data, collected by US-based air carriers between 2009





and 2014. Similarly, with respect to turbulence research as well as aviation applications (e.g., regional turbulence forecasting), statistics of a total of eleven EDR estimates (three wind-based EDRs for U, V, and W wind components and two DEVG-based EDRs) are investigated in the current study.

Figures 6 and 7 show the PDFs and lognormal curve fits of eleven EDR estimates from the B737 and B777 archived
data for 12 months from January to December 2012 (Fig. 1), respectively. Some of the lowest bins (open circle) are not used in the lognormal curve fitting, in order to obtain an optimized lognormal fit. At the lowest bins, the instrumental noise can affect some of the resulting EDR values, and nonturbulent conditions are of less interest than turbulent conditions. The curve fitting (line) is conducted using the Levenberg-Marquardt nonlinear least-squares fit. It is also noted that the data used in calculating the PDFs are binned with the samples with greater than 50 (15) reports in each bin among 50 bins for wind-based
EDRs (DEVG-based EDRs). Although some PDFs (e.g., Figs. 6g-i and Figs. 7a and 7g) tend to indicate a higher occurrence than the fitted lognormal fits in relatively lower bins, the lognormal curve fitting (continuous line of Figs. 6 and 7) shows good agreement with the PDFs (circle) at relatively higher bins for eleven EDRs of both the B737 and B777. The PDFs of the resultant EDRs computed using five different methods (a total of eleven EDRs) follow the lognormal distribution reasonably well, which satisfies the characteristics of atmospheric turbulence.

Table 1 shows the mean and SD of the natural logarithms of eleven EDRs [ln (EDR)] for each type of aircraft, which are obtained from lognormal curve fitting. Regarding the mean of ln (EDR), the B737 (from -4.75 to -2.81) is slightly larger than the B777 (from -6.32 to -2.97), and for the SD of ln (EDR), the B737 (from 0.54 to 1.17) is smaller than the B777 (from 0.62 to 1.72). For each aircraft type, the EDR estimates have similar statistics, except for EDR5. Compared to the EDR statistics (mean and SD) of previous studies [e.g., Sharman and Pearson (2017; -2.953 and 0.602, respectively), Sharman et al. (2014; -
2.85 and 0.57, respectively), and Kim et al. (2020; -2.94 and 0.63 for some of Asia, respectively)], the current EDR statistics have somewhat different values, except for EDR5 (-2.81 and 0.54 for B737, respectively, and -2.97 and 0.62 for B777, respectively), and some V-wind derived EDRs (EDR1V for B737 and EDR3V for B777). This discrepancy could be due to the relatively low sampling rate of the current data, differences in the spatiotemporal coverage of data, and aircraft type. Indeed, Kim et al. (2020) showed that these statistics can vary according to specified regions (Tables 1 and 2 of Kim et al., 2020).
Therefore, because the in situ equipped EDR data cover the Northern Hemisphere (e.g., Fig. 10 of Kim et al., 2020) and miss some of the turbulence observations between Asia and Europe, between Asia and Oceania, and within Asia, the statistics, such as the mean and SD, can be different according to region and season; this remains a future research topic of interest, which could be addressed by collecting sufficient aircraft measurements. As an alternative method to mitigate the imbalance of turbulence information globally, navigation information such as the ADS-B and Mode-S EHS can be applied to 1-Hz based
EDR estimation, together with 1-Hz aircraft measurements. Additional studies using more data and various types of aircraft data having relatively low sampling rate of aircraft data should be conducted, in order to obtain robust statistics on the observational characteristics of turbulence.



## 4 Results: Case analyses

For further evaluation of the derived EDRs from the 1-Hz aircraft data in this study, the EDR estimates from five methods are examined for selected strong turbulence cases. Strong turbulence events are determined based on the DEVG values, with a threshold of 4.5 m s$^{-1}$ for moderate-level turbulence (e.g., Truscott, 2000; Gill, 2014; Kim and Chun, 2016; Meneguz et al., 2016; Storer et al., 2019). Therefore, time series of eleven EDR estimates, three EDR1s (U, V, and W), three EDR2s (U, V, and W), three EDR3s (U, V, and W), EDR4, and EDR5 are examined.

### 4.1 Convectively induced turbulence (CIT) case

Figure 8 shows the flight route between Manila, Philippines and Incheon, South Korea (from 1613 to 1910 UTC 20 September 2012) and satellite images obtained from the infrared (IR) image of the Korean geostationary satellite, the Communication, Ocean, and Meteorological Satellite (COMS), at 1845, 1715, and 1745 UTC 20 September 2012. An aircraft heading for Incheon encountered a strong turbulence, with a change of vertical acceleration of more than 1 g at an altitude of ~37 kft near Taiwan over the Philippine Sea (121.64°E and 23.05°N; circle of Fig. 8) at 1715 UTC. Around the time of this turbulence encounter (from 1645 to 1745 UTC), locally isolated developing convective cloud was collocated at the region of the turbulence encounter. When the minimum IR brightness temperature ($T_b$) is calculated near the location of the turbulence encounter using 3-hourly GridSat-B1 data with a spatial resolution of 0.07° (Knapp et al., 2011), it is found that the minimum $T_b$ is the lowest at 1800 UTC which is the closest time with the turbulence encounter (not shown). This implies a rapid increase of cloud top height and corresponds well to the satellite images of Fig. 8. Therefore, we consider that this case is associated with turbulence above the rapidly developing isolated convection possibly with convectively induced gravity waves (e.g., Lane and Sharman, 2008; Kim and Chun, 2012; Kim et al., 2019b; Lane et al, 2003, 2012).

Figure 9 shows the time series of flight altitude, the DEVG, and eleven EDR estimates obtained from the 1-Hz QAR data on 20 September 2012. In Fig. 9a, a very strong and isolated peak of a large DEVG value (maximum value of 8.067 m s$^{-1}$) was found near the eastern side of Taiwan (circle of Fig. 8) at ~37 kft flight level. At that time, a high rate of change in altitude of 12.5–22 ft/s occurs (not shown). For this case, the time series of EDR estimates using each wind component (EDR1, 2, and 3) are examined separately (Figs. 9b-d). Note that the EDR4 and EDR5 in Figs. 9b-d are the same, because they do not use the wind data. A total of eleven EDRs exhibit the isolated peak and similar pattern to the vertical acceleration-based DEVG, although there are some differences in the EDR magnitudes. Among the wind-based EDRs (EDR1, 2, and 3), the EDR values estimated from the derived vertical wind velocity (W) are much larger than those from the zonal and meridional wind velocity (U and V). The largest values of EDRs derived from W (EDR = 0.739 m$^{2/3}$ s$^{-1}$) could be relevant to rapidly developing small-scale convection, which normally includes strong updrafts and flow deformation at the top of cloud, and generates subsequent convectively induced gravity waves above the convection with small-scale turbulent mixing near the top of convection (e.g., Lane and Sharman, 2008; Kim and Chun, 2012; Kim et al., 2019b; Lane et al., 2003, 2012). And, considering that the intensity criteria of light, moderate, and severe turbulence for mid-size aircraft such as the B737 are 0.15, 0.22, and 0.34 m$^{2/3}$ s$^{-1}$ of EDR


(Sharman and Pearson, 2017), the EDRs represent strong turbulence (severe turbulence), except for the EDRs derived from U (null or light intensity) and the EDR2 derived from V (moderate turbulence). The EDRs derived from W indicate that this was an extremely severe intensity and highly localized turbulence at the time that the turbulence event was observed.

## 4.2 Clear-air turbulence (CAT) case

Figure 10 shows two flight routes between Incheon and Seattle (route 1: from 0942 to 1836 UTC) and between Incheon and San Francisco (route 2: from 0830 to 1742 UTC) on 11 October 2012. Fig. 10 also shows the observed IR $T_b$ (Fig. 8a), 200 hPa and 250 hPa horizontal wind speed and vector (Figs. 8b and 8c, respectively), and 225 hPa vertical wind shear (Fig. 8d), computed using the European Centre for Medium-Range Weather Forecasts Re-Analysis, version 5 (ERA-5, Hersbach et al., 2020) reanalysis data with a horizontal grid spacing of 0.25° at 1200 UTC 11 October 2012. The observed $T_b$

is obtained from the GridSat-B1 data, with the horizontal grid spacing of 0.07° (Knapp et al., 2011). Two aircraft flying along both flight routes encountered strong turbulence with an abrupt change of vertical acceleration of ~ 0.74 g at an altitude between 35 and 37 kft, over the Northwestern Pacific Ocean (146.19°E and 37.46°N at 1125 UTC for route 1 and 145.59°E and 37.19°N at 1010 UTC for route 2). At that time, the intensified Typhoon Prapiroon was located in the Philippine Sea, which brought warm and moist southwesterly flows to the incident locations, while an intensified upper-level trough in the mid-latitude

provided a strong northwesterly, which brought increased meridional temperature gradients and provided favorable conditions for strong vertical wind shear via the thermal wind relationship with a high wind speed at the tropopause level over the turbulence regions (e.g., Kim and Chun, 2010, 2011; Williams and Joshi, 2013; Lee et al., 2019). Near the incident locations (shown as black open circles in Figs. 10b-c), the horizontal wind speed at 200 hPa (Fig. 10b) is stronger than 70 m s⁻¹, which is almost 25 m s⁻¹ higher than that at 250 hPa (Fig. 10c). This caused strong vertical shear of zonal winds between the 200 and

250 hPa levels in the upper-level jet (Fig. 10d), which results in shear instability to generate small-scale turbulence directly affecting cruising aircraft near the turbulence locations (e.g., Kim and Chun, 2010, 2011, 2016; Kim et al., 2011, 2018; Sharman and Lane, 2016; Storer et al., 2019). This case is considered as a conventional type of CAT due to the shear instability in the upper-level jet, although turbulence events were reported over the cloud, which seems to have a relatively broad $T_b$, like a cirrus cloud (Fig. 10a).

Figure 11 shows the time series in the same format as in Fig. 9, except for the CAT cases on 11 October 2012 (Figs. 10a-d for route 1 and Figs. 10e-h for route 2). At the time of turbulence occurrence, there are abrupt increases in the DEVG of 6.48 m s⁻¹ (route 1) and 10.75 m s⁻¹ (route 2) (Fig. 11a). At that time, the high rate of the altitude changed by more than 18 ft/s for both flight routes (not shown). Contrary to Fig. 9, with an isolated peak, the time series of DEVG for both routes 1 and 2 had more variations before and after the turbulence incident, especially route 1 (Fig. 11a). As found in Fig. 9, the EDR estimates

feature the strong CAT occurrence, and the temporal patterns of the EDR estimates and the DEVG are similar each other. However, the EDR values derived from U are larger than those from V and W. Considering turbulence cases located in the regions of the dominant upper-level jet stream, larger scale disturbance, such as a jet stream, can greatly affect turbulence generation (Cho and Lindborg, 2001), and this may lead to higher values of EDR from the zonal wind component than that





from other wind components. The EDR derived from W also has the second largest value, and this can be relevant to the spontaneous imbalance and emission of inertial gravity waves induced by the jet stream (Knox et al., 2008). Moreover, the EDRs indicate MOG-level turbulence (EDR $\geq 0.22$ m$^{2/3}$ s$^{-1}$), except for the EDR2 and EDR3 from V and EDR4. It is noteworthy that the EDR estimates from the V-wind component are significant and highly variant from different methods,

because this case is related to the Kelvin-Helmholtz billows due to strong shear instability which causes a strong *y*-component of vorticity (vortex tube; Clark et al., 2000; Kim and Chun, 2012).

### 4.3 Mountain wave turbulence (MWT) case

Figure 12 shows the flight route between Incheon and Toronto (from 0253 to 1507 UTC 30 December 2012) and the terrain height obtained using 5-minute digital elevation model data. Aircraft heading to Toronto Pearson international airport

encountered strong turbulence, with a change of the vertical acceleration of more than 1.3 g between 1012 and 1016 UTC at an altitude of ~33 kft over Alaska (148°-148.2°W and ~61.36°N), where low-level wind and upper-level wind jet streams existed (not shown). Therefore, the MWT case could be relevant to synoptic scale phenomena (Sharman and Pearson, 2017), and at the incident locations, the terrain height is locally steepened. This is clearly indicated in the zoomed field of Fig. 12. Indeed, Alaska has been considered as a representative mountain wave area (e.g., Sharman and Lane, 2016). In this regard,

although we need further investigation of the generation, propagation, and breaking of mountain waves in this case, this case can be related to mountain waves and their subsequent break down, which is one of the well-known turbulence sources (Kim and Chun, 2010, 2011, 2016; Sharman and Pearson, 2017; Kim et al., 2018).

Figure 13 shows the time series that are the same format as in Fig. 9, except for the MWT case on 30 December 2012. The DEVG recorded large values of more than 11 m s$^{-1}$ several times (Fig. 13a). As in Fig. 11, the DEVG shows more

variations before and after the peak than the CIT case. Among the three cases (CIT, CAT, and MWT cases), the MWT case had the largest variation of vertical acceleration, and the largest magnitude of the resultant DEVG. For four minutes, the aircraft collected eight MOG-level (four severe and four moderate) turbulence reports. This implies that the scale of sources (synoptic-scale) for CAT and MWT is different from that for CIT (isolated mesoscale convective cell). The DEVG-based EDRs are more than 0.7 m s$^{-1}$, which corresponds to severe turbulence based on Sharman and Pearson (2017). As shown in

Figs. 9 and 11, the EDR estimates capture the MWT occurrences well, and the patterns of EDR estimates are similar to each other. However, in the MWT case, the EDR derived from W is larger than that from U and V. As the aircraft flew above the mountainous regions, vertically propagating gravity waves may have perturbed the background conditions, and lead to an environment conducive to turbulence generation. Furthermore, mountain-wave amplification and its subsequent breaking lead to small-scale turbulent mixing directly. A bumpy ride caused by mountain waves can be related to the large values of

the EDR estimates from both the U and W wind components. Like Fig. 11, the EDRs indicate severe turbulence (EDR $\geq 0.34$ m$^{2/3}$ s$^{-1}$), except for the EDR2 and EDR3 from V. The current study examines the feasibility of using 1-Hz EDR estimates for strong (MOG-level) turbulence cases that are related to CIT, CAT, and MWT. We found that a total of eleven EDR estimates capture well the turbulence cases generated by different mechanisms, in terms of both their intensity and temporal





patterns. It is also noteworthy that the characteristics of EDR estimates that aircraft feels can vary depending on various sources of turbulence. As far as we know, there is no work done to compare the accuracy of the EDR observations with synoptic or mesoscale regimes. More intercomparison of various objective EDR estimations from aircraft data with respect to the possible sources is necessary in the future.

## 5 Summary and discussion

In the current study, we derive the EDR using a relatively low sampling frequency (1 Hz) of aircraft measurements compared with a relatively high sampling frequency (e.g., 8 Hz or 10 Hz) of in situ EDR measurements. We use the retrieved 1-Hz QAR data of the B737 and B777 for 12 months (January to December 2012) obtained from KAL post-flight recorders. These 1-Hz post-retrieved flight data are cost effective with respect to both data link and storage compared to a relative high sampling frequency (e.g., 8 Hz or 10 Hz) of in situ EDR measurements. The wind (zonal, meridional, and vertical wind) information and the DEVG are used to compute the EDR separately. The vertical wind data that were not included in the QAR dataset are estimated using some flight parameters, such as the angle of attack, inertial vertical velocity, and roll angle. Three wind components are used to calculate three different EDRs, utilizing the structure function, PSD, von Kármán wind model, and maximum likelihood method-based EDR estimation, under the assumption of Taylor's frozen hypothesis. Lastly, two DEVG-based EDRs are computed using the prescribed parabolic curve proposed by Kim et al. (2017) and lognormal mapping technique proposed by Sharman and Pearson (2017).

Using these five methods (three wind-based and two DEVG-based methods), the eleven EDR estimates are computed for each type of aircraft, and the results are tested to objectively measure the feasibility of turbulence detections associated with various sources of turbulence events. The findings are summarized as follows:

1) It is found that 1-Hz EDR estimates exhibit good agreement with selected MOG-level turbulence events, with respect to turbulence intensity and temporal patterns that are related to CAT, MWT, and CIT, with different characteristics of the observed EDRs.

2) Zonal (vertical) wind-based EDRs are stronger in the CAT (CIT) case, while MWT has the peak of EDRs in both zonal and vertical wind-based EDRs.

3) The current EDR estimates from five different methods follow a lognormal distribution reasonably well, which satisfies the fundamental characteristics of atmospheric turbulence.

4) The statistics (mean and standard deviation) of log-scale EDRs are somewhat different from those of a previous study using a higher frequency (e.g., 8 Hz or 10 Hz) of in situ aircraft data in the US, likely due to different sampling rates, aircraft types, and locations.

This suggests that the EDR estimates, using a relatively low sampling rate of flight data such as the current 1-Hz data, could be useful for turbulence reporting, and could eventually provide a more widespread database of aviation turbulence observations. The results in the case studies also suggest that characteristics of the observed EDR estimates using U, V, and



W are different from the possible sources of atmospheric turbulence in the cruising altitudes (Upper Troposphere and Lower Stratosphere, UTLS. This is important because it is useful for classifying the possible sources of turbulence in the UTLS from the aircraft-based observation (ABO) data by itself, which can eventually provide a basic information for nowcast (tactical turbulence avoidance) and can be useful for classification of the objective evaluations of turbulence forecast systems (e.g.,

Sharman et al., 2006; Kim et al., 2011, 2018, 2019a, 2021c; Kim and Chun, 2016; Sharman and Pearson, 2017).

For nowcasting, the CAT and MWT cases occurred by large-scale (synoptic-scale) forcing have longer variations in the EDRs before and after the turbulence incident, while the CIT case triggered by a smaller scale mesoscale convective cell has an isolated (highly localized) peak of EDR. This feature could be useful for pilots to take more proactive action to turn on the seat belt sign before the CAT and MWT are expected to happen. If they consider that CAT is more likely to be concentrated

in a shallow layer above or below the jet core (like a pancake), this could be also useful for them to make decisions to avoid these areas by only changing altitude for CAT cases with strong U-wind variations in the on-board parameters.

From the turbulence prediction perspective, up to now we have three different types of turbulence forecasts: CAT, MWT, and CIT (e.g., Sharman et al., 2006; Kim et al., 2011, 2018, 2019a, 2021c; Kim and Chun, 2016; Sharman and Pearson, 2017). To evaluate these forecasting systems, we need more organized database of the observed EDR estimations that should

be classified into three possible sources. This study is the first attempt to understand the characteristics of possible sources of the EDR observations from the 1-Hz ABO data by itself. Here, the basic assumptions are listed as below:

1)    CAT is affected by large-scale forcing such as upper-level jet/frontal system that has strong variation in zonal wind (e.g., Dutton and Panofsky, 1970; Ellrod and Knapp, 1992; Kim and Chun, 2010) and geostrophic imbalance with emissions of inertial gravity waves with the larger horizontal wavelengths (e.g., Lane et al., 2004; Zhang et al., 2004; Koch et al.,

2005; Knox et al., 2008; Ellrod and Knox, 2010).

2)    MWT has the large variations in both zonal and vertical wind in this study. There are two reasons for this. First, vertically propagating mountain waves are highly driven by large-scale flows across the mountain (e.g., Lane et al., 2009). Second, large amplitudes of mountain waves and their subsequent breaking are revealed by strong magnitudes of vertical velocity fields (e.g., Kim and Chun, 2010; Sharman et al., 2012).

3)    For CIT, convectively induced gravity waves outside the cloud boundary (e.g., Chun and Baik, 1998; Lane et al., 2003; Kim et al., 2019b, 2021c) or strong updraft/downdraft inside the cloud (e.g., Lane et al., 2003; Kim and Chun, 2012) have highly transient and localized features with strong vertical velocity. Therefore, CAT, MWT, and CIT cases in this study have different characteristics in the EDR estimations based on U, V, and W.

To summarize, although more case analyses need to be required in the future, current results can be a useful reference

when pilot strategize flight planning in real-time perspective and can be fundamental information to evaluate the forecasting systems separately to the sources.

Considering the complicated process required to collect aircraft measurements via collaboration with commercial airlines, this will be eventually useful for applying these EDR estimation methods to the 1-Hz real time data in future. Possible candidates for the real-time data are the navigational information of commercial aircraft such as the ADS-B and Mode-S EHS.



Given that the set-up for the ADS-B or Mode-S EHS receiving stations does not require a lot of work or money for transmission of the data, EDR estimates from ADS-B and Mode-S EHS (e.g., Kopeć et al., 2016) could greatly support the construction of a turbulence database and statistics globally, together with on-board based (both fine and coarse) EDR measurements (e.g., Sharman et al., 2014; Gill, 2014) and other sources of data like radiosonde, weather radars, and lidars (e.g., Bodini et al., 2019; Ko et al., 2019; Kim et al., 2021a, 2021b). We leave the real time experiment as a future study, because it requires a huge amount of additional works to set up the receivers, to conduct the real-time quality control, and so on (e.g., de Haan, 2011; Stone and Pearce, 2016). From the ADS-B and Mode-S, it will be expected that the 1-Hz real-time turbulence information (e.g., Krozel and Sharman, 2015; Kopeć et al., 2016) can be useful for constructing the real-time flight strategy considering sources of turbulence in future.

*Author contributions.* SHK, JHK, and JK designed the study. SHK prepared the original draft of the paper, with contributions from JHK, JK, and HYC. Together, SHK, JHK, JK, and HYC interpreted the results and reviewed and edited the paper.

*Competing interests.* The authors declare they have no conflict of interest.

*Acknowledgements.* This work was funded by the Korea Meteorological Administration Research and Development Program under Grant KMI2020-01910, and was supported by the Basic Science Research Program through the National Research Foundation of Korea (NRF) funded by the Ministry of Education (NRF-2019R111A2A01060035).

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



**Table 1.** Values of the mean and standard deviation (SD) of the natural logarithms of EDRs computed from (a) B737 and (b) B777 QAR datasets.

(a) B737

|  | EDR1U | EDR2U | EDR3U | EDR1V | EDR2V | EDR3V | EDR1W | EDR2W | EDR3W | EDR4 | EDR5 |
|---|---|---|---|---|---|---|---|---|---|---|---|
| Mean | -4.1021 | -4.5715 | -4.2185 | -3.9484 | -4.6723 | -4.3012 | -4.7057 | -4.0320 | -3.6601 | -4.7517 | -2.8100 |
| SD | 0.7380 | 0.8224 | 0.8218 | 0.7284 | 0.7848 | 0.7831 | 1.1744 | 0.9739 | 0.9706 | 0.7519 | 0.5441 |

(b) B777

|  | EDR1U | EDR2U | EDR3U | EDR1V | EDR2V | EDR3V | EDR1W | EDR2W | EDR3W | EDR4 | EDR5 |
|---|---|---|---|---|---|---|---|---|---|---|---|
| Mean | -6.0813 | -4.5119 | -4.1517 | -4.2144 | -4.3372 | -3.9596 | -6.3149 | -5.0176 | -4.6488 | -5.3378 | -2.9723 |
| SD | 1.6361 | 1.0995 | 1.0970 | 0.9101 | 0.8414 | 0.8401 | 1.7153 | 1.2803 | 1.2830 | 0.9724 | 0.6231 |



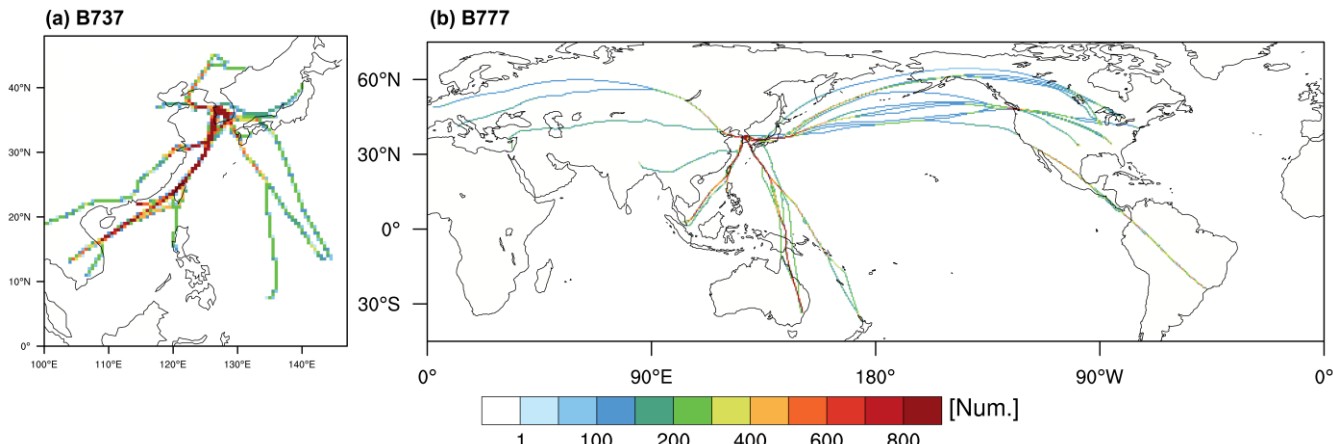

**Figure 1.** The horizontal distribution of the number of (a) B737 and (b) B777 aircraft data at altitudes above 15 kft, accumulated within a 0.5°×0.5° horizontal grid box for the 12 months from January to December 2012.



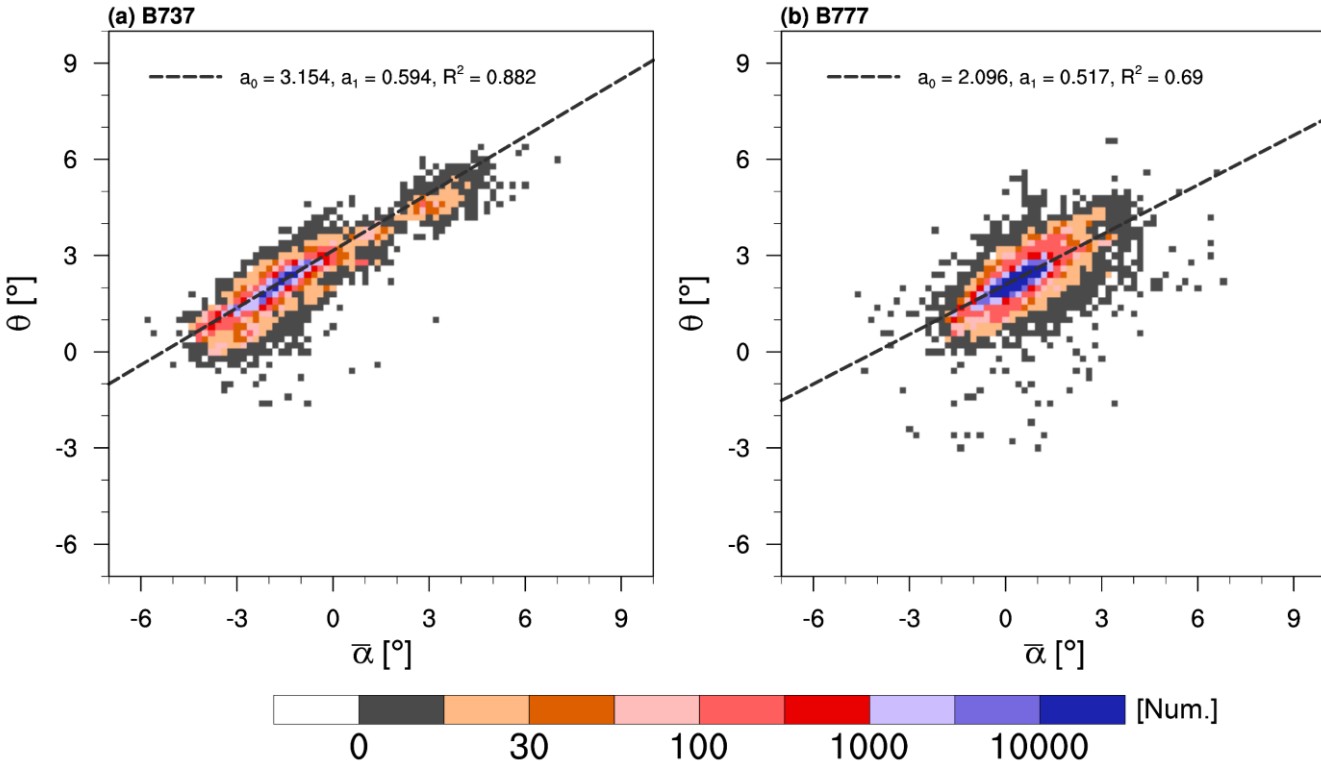

**Figure 2.** Scatter density plots (circle) of the measured angle of attack ($\bar{\alpha}$) and pitch angle ($\theta$), along with the least-squares linear regression fits (dashed line), for the (a) B737 and (b) B777. The least-squares intercept, slope, and degree of goodness of fit are written as $a_0$, $a_1$, and $R^2$, respectively.





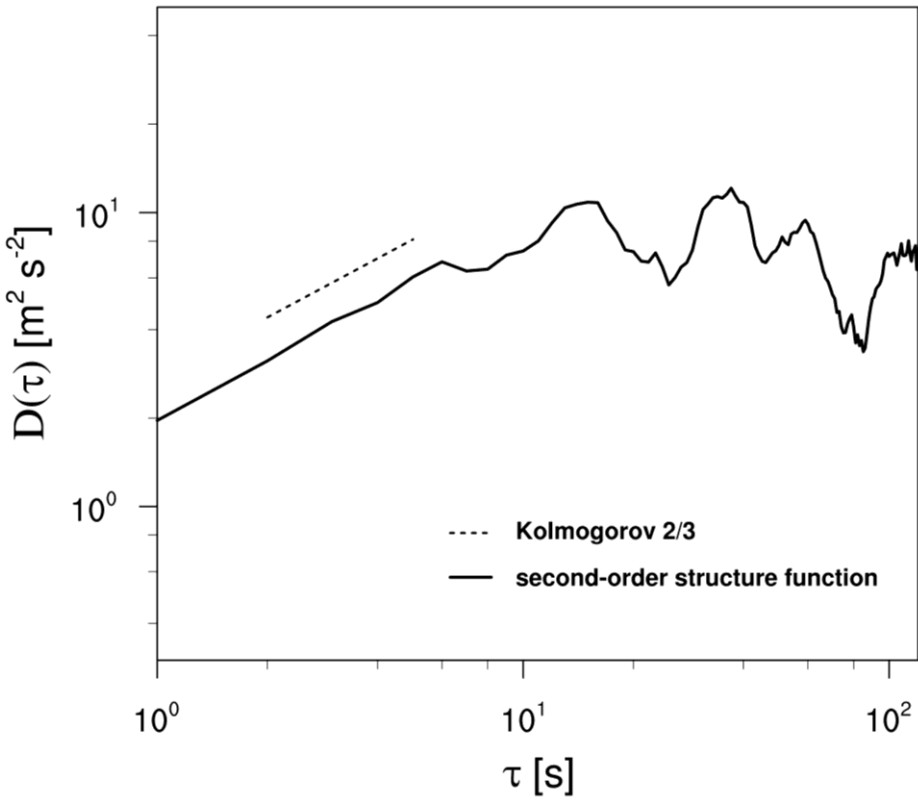

**Figure 3.** Example of the second-order structure functions of the meridional wind component obtained from the QAR data for 2 minutes (starting from 1128 UTC 11 October 2012) when strong turbulence (DEVG = 5.555 m s$^{-1}$) was observed. The dashed line represents the theoretical Kolmogorov's inertial range slope $\tau^{2/3}$ in the time domain.



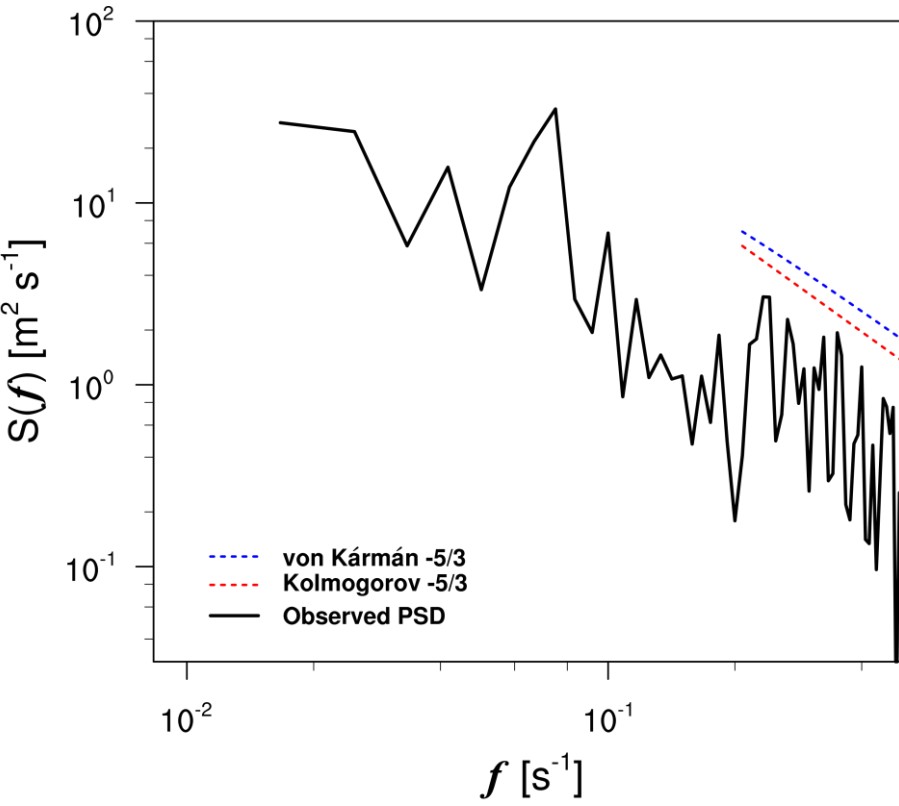

**Figure 4.** The energy spectrum of the zonal wind component obtained from the QAR data at 1038 UTC 11 October 2012. The dashed line represents the theoretical von Kármán (blue) and Kolmogorov's (red) inertial range slope $f^{5/3}$ in the frequency domain.

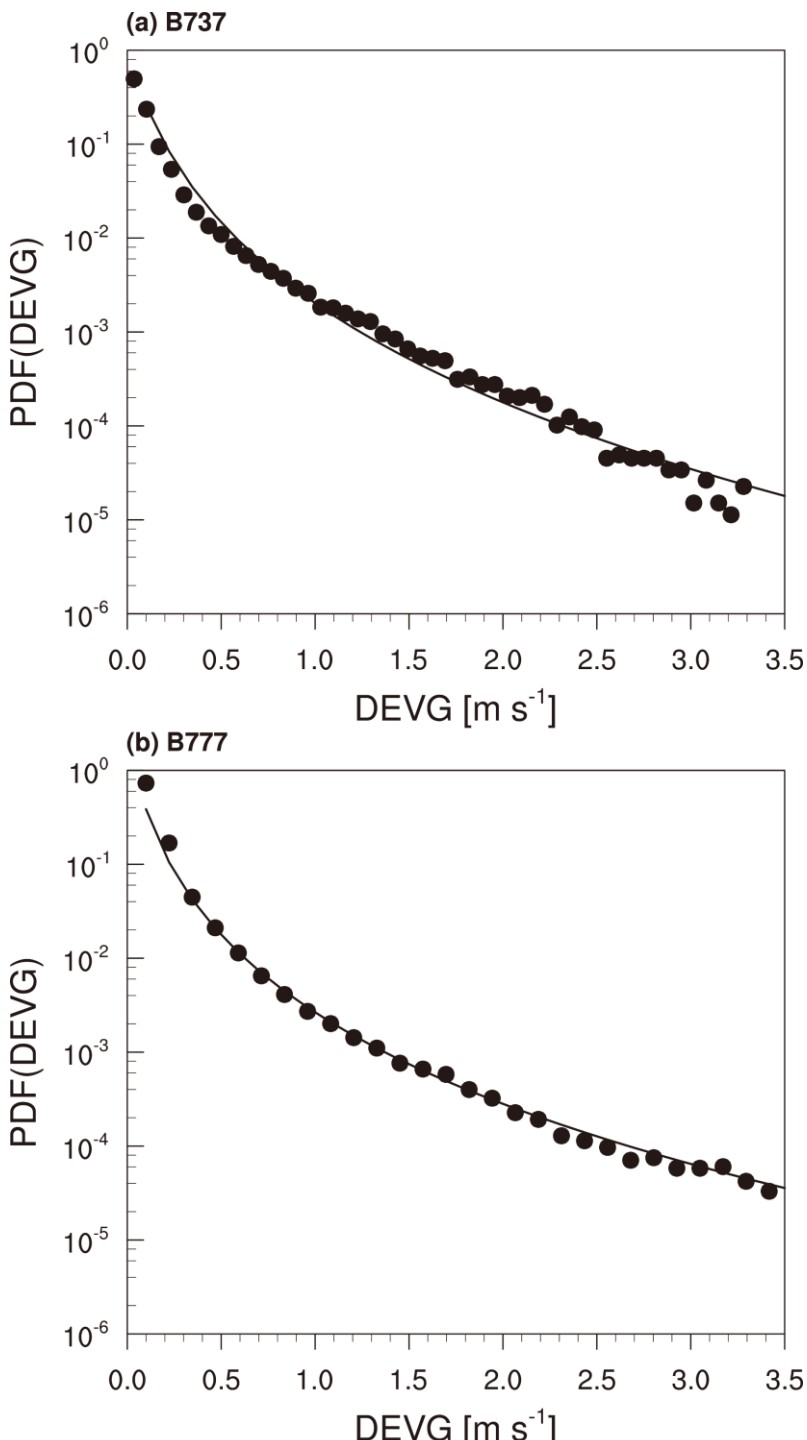

**Figure 5.** The probability density functions (PDFs) of the DEVG and lognormal fits (line) over the DEVG for the (a) B737 and (b) B777 accumulated for the 12 months from January to December 2012.





**Figure 6.** The PDFs (circle) of the EDRs and lognormal fit (continuous line) over the EDRs obtained from the B737 data. The

filled circles indicate data that were used in the fit, and the open circles indicate data that are excluded from the fit.





**Figure 7.** As in Fig. 6, but for B777.



**Figure 8.** (a) The flight route (line) from 1613 to 1910 UTC 20 September, IR image obtained from the COMS at (b) 1715 UTC when the turbulence was encountered, (c) 1645 UTC before the incident time, and (d) 1745 UTC after the incident time. The horizontal location of the turbulence encounter is represented by a circle.





**Figure 9.** Time series of (a) flight altitude and DEVG and (b-d) EDR estimates obtained from the QAR data on 20 September 2012. The maximum value of the EDR is written in parentheses.

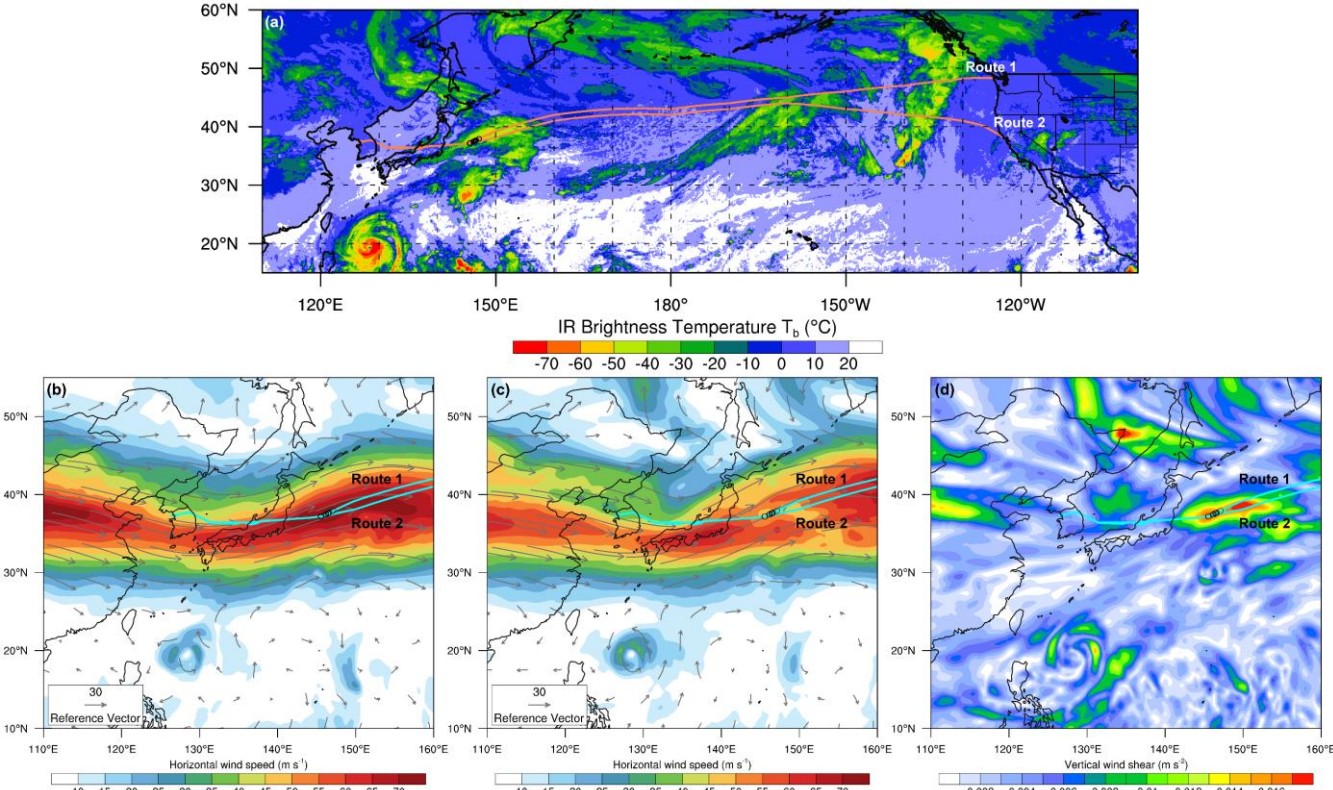

**Figure 10.** (a) Observed infrared brightness temperature and (b, c) horizontal wind speed (shading) and wind vector (curly arrow) at 200 hPa and 250 hPa, respectively, and (d) vertical wind shear at 225 hPa computed from the ERA-5 reanalysis data at 1200 UTC 11 October 2012. The flight routes (line) and horizontal locations (circle) of MOG-level turbulence events from two QAR datasets (routes 1 and 2: from 0942 to 1836 UTC and from 0830 to 1742 UTC, respectively) are superimposed.



**[Route 1]**

**[Route 2]**

**Figure 11.** As in Fig. 9, but for (a-d) route 1 and (e-h) route 2 of the QAR data on 11 October 2012.

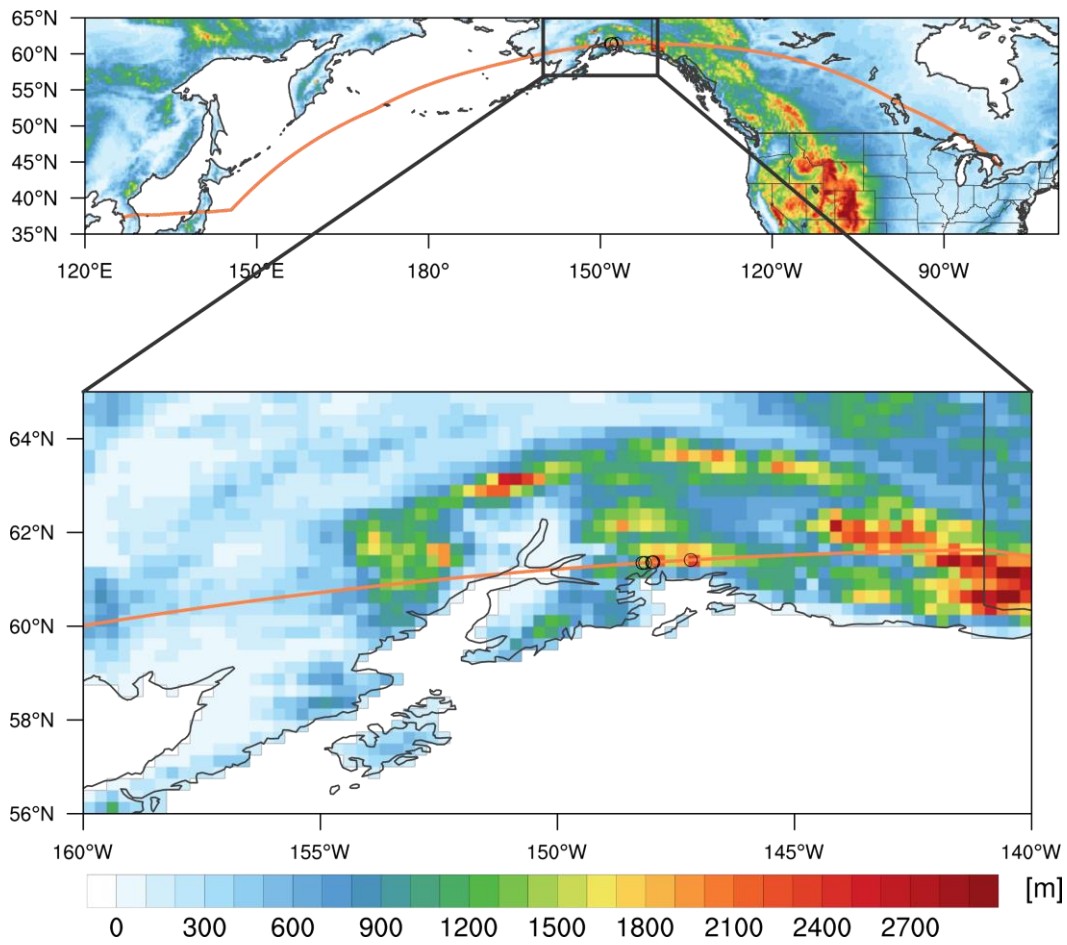

**Figure 12.** Flight route (line) and horizontal locations (circle) of MOG-level turbulence events detected from the QAR data on December 2012 (from 0253 to 1507 UTC 30 December), with a horizontal distribution of terrain height of 5-minute digital elevation model data.





**Figure 13.** As in Fig. 9, but for the QAR data on 30 December 2012.