# Peer review of "Characteristics of the Derived Energy Dissipation Rate using the 1-Hz Commercial Aircraft Quick Access Recorder (QAR) Data"

_Atmospheric Measurement Techniques, 2021_

## Referee Comment (RC1)

**Characteristics of the Derived Energy Dissipation Rate using the 1-Hz Commercial Aircraft Quick Access Recorder (QAR) Data**

*Soo-Hyun Kim , Jeonghoe Kim , Jung-Hoon Kim , and Hye-Yeong Chun*

**REVIEW**

**GENERAL COMMENT:**

The paper provides a nice and detailed analysis of energy dissipation rate (EDR) calculated with several approaches using 1 year of data collected by two commercial aircrafts. The paper is well written, the figures generally well made (but please see my comments below), and the analysis is explained in detail. While this reviewer is not an expert in aviation meteorology, I am sure the topic is of great interest for the community. My review focused on the turbulence calculations and boundary layer meteorology aspects of the analysis, and from that point of view I find the paper worth of being accepted after minor revisions.

**MAJOR COMMENTS:**

1. Most of the abstract explains the methodology applied to derive EDR, which are not novel or unique to this analysis. I recommend re-shaping the abstract so that more focus is given to the results of this analysis, and especially its novelty aspects compared to previous studies in the same field.
2. P. 6 l. 22: how did you select the "predefined inertial range"? Was this selection valid at both high and low turbulence regimes? At all heights? For both aircrafts? This is critical choice that impacts the EDR calculation, and therefore should be explained in more detail.

**SPECIFIC COMMENTS:**

1. P.2 L.22: should be "Southwest". Also, these references to commercial airlines probably require a citation.
2. P.4 L.5: The following sentence is confusing and somewhat misleading; you should just say that the B737 data were recorded at 0.25 Hz. The same type of misleading information is used many times in the rest of the paper. "The aircraft flight parameters used in the current 5 study were recorded every second (1 Hz) for both the B737 and B777. Because the 1-Hz wind direction and wind speed of the B737 records had the same values within a 4-second time window, the wind direction and wind speed of the B737 records seem to have 1/4-Hz sampling frequency."
3. P. 6 L. 3: please delete the following sentence as you have already stated this piece of information: "(264,867 and 1,065,855 reports from B737 and B777 recorders, respectively)".

4. Eq. 5: please define what the brackets "<>" mean.
5. P. 15 L.2: missing ")".
6. P.16 L. 6: "and so on" is not appropriate for a scientific paper, please rephrase.
7. Figures 9, 11, 13: can you consider using a log scale on the y-axis of panels b, c, and d? Right now, the plots are very hard to read.
8. Figure 12: it's hard to see the black circle(s), please use a different color.
9. Throughout the paper, please consider using a different color scheme. The rainbow color has quite a lot of issues, see for example https://www.climate-lab-book.ac.uk/2014/end-of-the-rainbow/

---

## Author Comment (AC3)

**[ Responses to the Comment by the Anonymous Referee #4 ]**

>> We deeply appreciate the referee #4 for providing constructive comments. The manuscript is revised following the comments below.

This article investigates the applicability of Quick Access Recorder (QAR) in-situ aircraft data obtained at a 1 Hz sampling rate as a cost-effective alternative to derive energy dissipation rates (EDR). The authors compare 5 methods to estimate EDR using QAR data collected during 2012 from Korean-based air carriers (B737 and B777 aircraft). The study is overall well organized and written. Plus, the topic itself it is certainly of great value to the aviation industry and the turbulence research community at large. However, there are a number of major aspects that need to be addressed upon publication at the journal of Atmospheric Measurement Techniques. Given the magnitude of these issues, it may be more appropriate for the authors to consider resubmission in the future. Here below I provide both general and specific comments with the hope that these can help the authors get to a publication stage.

**General comments:**

1) Most of manuscript is devoted to the description of the different methods used to derive EDR. While this is of value, none of these approaches are firstly introduced herein, so there needs to be significantly more weighting in the analysis and quantification of the usability of QAR (see other comments below).

➔ Thank you very much for the good comment. We reflected the reviewer's general comments in the revised manuscript. Also, the abstract is also reshaped to emphasize the analyses on characteristics of the EDR estimates. [Page 1, Lines 7-27]

2) A main drawback of the current study is the lack of higher-rate data to use as reference in the estimation of EDR. Even if the 5 methods produce similar EDRs, all estimates can be biased. It is impossible to use such comparison in the lack of a higher-accuracy reference (e.g., 10 Hz data) to determine the adequacy of 1 Hz QAR-derived EDR estimates. Otherwise, one can only comment on the differences between the 5 methods, which has already been reported for most of them in the literature, and that is not the aim of the current manuscript [Page 3, lines 21-22: "The main purpose of this study is to examine the feasibility of various objective EDR estimations using the 1-Hz (coarser frequency) flight data for possible sources of atmospheric turbulence in cruising altitudes"]. If simultaneous retrievals of 10 Hz and QAR are not available,

I would suggest to use 10 Hz data decimated to resemble the 1 Hz QAR. This will lead to a fair assessment of the QAR data and that will help understand any systematic biases or errors at a particular range of EDR.

➔ Thank you very much for the good comment. Unfortunately, we do not have any higher-frequency (e.g., 10-Hz) QAR data. Therefore, we applied our method to the 20-Hz wind data obtained from the Boseong Meteorological Observatory (BMO), South Korea for 10 days (between 0000 LST 1 October and 0000 LST 10 October 2019). For a direct comparison, raw 20-Hz wind data are subsampled to the 1-Hz wind data using Reynolds averaging. It is noted that Kim et al. (2021) derived the energy dissipation rate (EDR) using the BMO high-frequency sonic data. In general, 1-Hz data shows similar distribution with the 20-Hz data although some peaks revealed in the 20-Hz wind data are smoothed in the 1-Hz wind data (Fig. A1). In the energy spectrum of streamwise component of 20-Hz and 1-Hz BMO wind (U-wind) data at 300 m above ground level at a certain time (Fig. A2), both spectra show a similar distribution, following a theoretical slope (-5/3) especially in the overlapped range of frequency (0.1-0.5 s$^{-1}$). Although two spectra have minor discrepancy in the overlapped frequency range of 0.1-0.5 s$^{-1}$, we confirmed that 1-Hz data have quite similar energy spectra with higher-rate data.

[Figure]

**Figure A1.** Timeseries of streamwise component of wind data (U) obtained from the BMO between 1000 LST and 1030 LST 1 October 2019 at 300 m AGL. Red and green lines indicate raw 20-Hz wind data and mean 1-Hz wind data, respectively.

[Figure]

**Figure A2.** The energy spectrum of U-wind obtained from raw 20-Hz (black) and subsampled 1-Hz (blue) BMO data at 1025 LST 1 October 2019 at 300 m AGL. The dashed line represents the theoretical slope ($f^{5/3}$) in the frequency domain.

We also calculated the EDR based on three inertial dissipation method (EDR1, EDR2, and EDR3) using the zonal wind of 20-Hz and 1-Hz BMO data for 10 days (between 0000 LST 1 October and 0000 LST 10 October 2019). When the timeseries of EDRs calculated using two datasets are examined, variations in the EDRs are quite similar in 20-Hz and 1-Hz BMO data although there are some underestimation or overestimation in the magnitude of the EDR (not shown). In scatter density plots of 20-Hz EDR and 1-Hz EDRs from the BMO datasets (Fig. A3), it is found that 1-Hz EDR data is highly correlated with 20-Hz EDR data (r > 0.92), implying that the 1-Hz data can provide a reliable information (timing and location) of atmospheric turbulence. Based on this result, we can confirm that the use of 1-Hz flight data to estimate the EDR is reliable, although the magnitude of the EDR is somewhat underestimated in some cases. Intercomparison between 1-Hz and higher frequency aircraft data will also be conducted in the future. The abovementioned statements are included and Figs. A2 and A3 are included as a new figure (Figure 5) in the revised manuscript. [Page 8, Lines 14-28]

During the revision process, the statements on a comparison of EDR climatology between the higher-rate (e.g., 10-Hz or 20-Hz) data and 1-Hz QAR data are deleted in the revised manuscript. In the revised manuscript, we include the statements that 1-Hz QAR data can be an additional source for measuring EDR when the high-frequency in-situ aircraft measurement is not available and this can contribute to expand more EDR information at the cruising

altitudes in the world. [Page 3, Lines 20-24; Page 15, Lines 17-19]

[Figure]

**Figure A3.** Scatter density plots of the EDRs calculated using the 20-Hz wind data (x-axis) and the 1-Hz wind data (y-axis) obtained from the 10-day BMO data. Pearson correlation and MAE are given in the top-left corner of each panel.

3) The use of a climatological log-normal distribution as reference may not be adequate in this case neither is supported by the presented results. There are reasons why the PDFs of EDR do not behave as a log-normal distribution. These can be related to the limited data availability (seasonal effects) and/or geographical considerations. To me, a higher probability of null-light turbulence that decreases as EDR increases makes good sense, as turbulence encounters are a rare event in a climatological sense. Therefore, the assumption that the QAR-derived PDF of EDR has to be log-normal is not necessarily a good way to assess the performance of this lower-frequency data.

➔ Thank you very much for pointing out such an important issue, which we did not properly mention in the original manuscript. We did not intend that the lognormality is used for

demonstrating the reliability of data in the original manuscript. Instead, we intended to report characteristics revealed in the QAR-derived probability density functions (PDFs) in the original manuscript. To avoid any confusion, the statement is modified in the revised manuscript. [Page 10, Lines 32-33; Page 11, Lines 1-2; Page 15, Lines 12-13]

3-1) Page 10, Lines 5-6: This is questionable. In the majority of cases there is not a reasonable fit for the EDR values at the peak and below, so I do not think these types of distributions are log-normal. Also, the higher probability bins should have a larger impact on the fit, so the fact that some of these needed to be excluded appears to be the result of a clear departure from a log-normal distribution. Maybe there is some issue with the particular optimization method selected (Levenberg-Marquardt) or its implementation by the authors. Perhaps trying another least-square minimization algorithm will help understanding where the actual problem comes from. Some level of discrepancy among the PDF and the best is to be expected, specially toward the tails of the distribution, but not for the area of larger probability if the selected reference is good descriptor of the observed quantity.

➔ Thank you for the good comment. During the revision process, we conducted lognormal fit obtained using other optimization methods [Powell's method (Press et al. 1992) and Nelder-Mead method (Gao and Han 2012)] to the EDR (Figures A4b and A4c, respectively). Compared to the Levenberg-Marquardt method used in the original manuscript (Fig. A4a), mean and standard deviation of the natural logarithm of the EDR (EDR1V) have similar value depending on each optimization method ($C_1$ and $C_2$ in Fig. A4, respectively) and this can also be found in all five EDR estimation methods (not shown). In this regard, it is considered that the choice of minimization algorithm will not have a significant impact on the resultant lognormal fit. These statements are included in the revised manuscript. [Page 10, Lines 30-32]

[Figure]

**Figure A4.** The PDFs (circles) of the EDR calculated using meridional wind (EDR1V) and lognormal fit (dashed line) over the EDRs obtained from the B777 data. The filled circles indicate data that were used in the fit, and the open circles indicate data that are excluded from the fit. The lognormal fitting is conducted via (a) Levenberg-Marquardt algorithm (used in the original manuscript), (b) Powell's method, and (c) Nelder-Mead method. The mean and standard deviation of the natural logarithm are written in each figure ($C_1$ and $C_2$, respectively).

3-2) Page 10, Lines 11-14: This claim is not supported by the data. I agree that the interest is more on larger EDR values, but the presented results do not justify the claim of a 'reasonable' log-normal distribution. This seems to be a key point of the manuscript to prove the validity of the QAR-derived EDR estimates, which makes me wonder about the value of these study lacking a reference (ideally 10 Hz data) to compare to.

➔ Thank you for pointing out this. As responded above, we included the validity of the 1-Hz EDR estimates using the BMO data in the revised manuscript [Page 8, Lines 14-28]. The statement related to the lognormal distribution is modified in the revised manuscript. [Page 10, Lines 32-33; Page 11, Lines 1-2]

3-3) Page 10, Line 18: Having similar statistics (mean and std) does not guarantee any skill in any of them (see general comment #1).

➔ Thank you for the comment. As responded in your comment 3), we did not intend that the

lognormality is used for demonstrating the reliability of data in the original manuscript. The statements related to the PDFs are modified in the revised manuscript [Page 10, Lines 32-33; Page 11, Lines 1-2]. Also, as responded above, we applied our method to the 20-Hz BMO data and subsampled 1-Hz BMO data and it is shown that 1-Hz EDR data is highly correlated with 20-Hz EDR data, although the magnitude of the EDR can be underestimated in some cases. These statements are also included in the revised manuscript. [Page 8, Lines 14-28].

4) The 'Results: Case analysis' section is highly subjective and does not allow to quantitatively assess the QAR-based EDR estimates. These are interesting cases, including the discussion of the associated potential large-scale mechanisms leading to turbulence. However, without a reference EDR to compare to, they do not add much to the verification and establishment of QAR data as a reliable source of data to derive EDR, neither to the actual strengths/downsides of each of the 5 exercised approaches to estimate EDR.

➔ Thank you for the good comment. As responded above, we indirectly tested the validity of the 1-Hz EDR by applying the same methods to both higher frequency (20-Hz) raw data and subsampled 1-Hz data from the BMO data. Statements related to this are included in the revised manuscript [Page 8, Lines 14-28]. Based on this result, it can be confirmed that the use of 1-Hz flight data to estimate the EDR is comparable. Therefore, we kept 'Results: Case analysis' section in the revised manuscript.

5) There are some details regarding the application of some of the techniques used to derive EDR that need attention, including the use of B737 data at 0.25 Hz. See the specific comments down below.

➔ Thank you very much for the good comments and suggestions. We addressed the specific comments below. Regarding to the original statement of the use of Boeing (B) 737 data, both the B737 and B777 data are recorded every second. To avoid any confusion, we modified the original sentence in the revised manuscript. [Page 4, Lines 4-5]

**Specific comments:**

1) Page 1, Lines 18-20: This statement is not supported by the results (See general comment #3).

➔ During the revision process, this statement is deleted.

2) Page 1, Lines 20-22: Or maybe inappropriate range of the spectrum being captured. This is just a speculation that needs further support.

➔ During the revision process, this statement is deleted.

3) Page 2, Line 24: What does the term 'off-line' refers to here? Post-flight? Please explain.

➔ The term 'off-line' in the original manuscript is changed into 'post-flight' in the revised manuscript. [Page 2, Line 26]

4) Page 3, Lines 13-14: What are the advantages/disadvantages of using QAR versus EHS and ADS-B? It would be good to put QAR in the context of data quality, availability and access compared to these other 1Hz sources, as well as elaborate on the fundamental principles and assumptions used to derive EDR in each case. Maybe this discussion is more appropriate in the methods section.

➔ We are sorry if the statement of the original manuscript brought a confusion. The main purpose of this study is not comparing the QAR data with Mode-S Enhanced Surveillance (Mode-S EHS) and Automatic Dependent Surveillance-Broadcast (ADS-B) data, but trying to find out an additional source of EDR estimations by applying all possible EDR methods to the currently available sources of low-frequency data (1-Hz QAR data in this study). To avoid any confusion, the sentence is modified in the revised manuscript. [Page 3, Lines 24-25]

5) Page 3, Lines 26-27: I would suggest removing this sentence from here since it is mentioned in the methods section.

➔ The sentence is deleted as suggested.

6) Page 3, Line 30: "MOG-level turbulent cases". Based on what? It should be an independent method… PIREPs?

➔ The moderate-or-greater (MOG)-level turbulence cases were determined based on derived equivalent vertical gust (DEVG $\geq$ 4.5 m s$^{-1}$, Gill 2014). The statement is included in the revised manuscript. [Page 3, Line 31]

7) Page 6, Line 22: This may be a bit too large of a scale for the aircraft to feel since wind span is less than 100 m (~50 m).

➔ Thank you for the comment. Yes, we partly agree with your opinion. Given the lowfrequency (1-Hz) data, we chose predefined inertial range that has the minimum size of horizontal wavelengths. In addition to that, we found the inertial range that minimizes errors between theoretical slope and observed slope. This statement is included in the revised manuscript. [Page 6, Lines 23-26]

8) Page 6, Line 28: Should mention that the range is 4-5s for the B737.

➔ We are sorry for if the statement of the original manuscript brought a confusion. Given that the recording frequency of the B737 data is also 1-Hz, the same inertial range is applied. The related statement is modified in the revised manuscript. [Page 4, Lines 4-5]

9) Page 7, Line 1: How about the 0.25 Hz?

➔ We are sorry again for if the statement of the original manuscript brought a confusion. The recording frequency of the B737 data is also 1-Hz. The related statement is modified in the revised manuscript. [Page 4, Lines 4-5]

10) Page 7, Line 10: "no window". This seems contradictory. The next sentence indicates that a window has been applied. Do the authors mean 'no windowing'? If so, a 2-min FFT can be rather noisy (as clearly evident Fig. 4), tending to over-estimate EDR (e.g., Munoz-Esparza et al. MWR2018 in the references). The same would apply to method 3 (von Karman spectrum). Please comment on this.

➔ Thank you for pointing out this. We conducted a fast Fourier transform using a Welch window with no overlap. The original term "no window" is changed into "no overlap" in the revised manuscript. [Page 7, Line 12]

11) Page 7, Lines 16-17: The von Karman model is meant to capture larger scales, so it seems like the authors would need to consider a wider frequency range than 0.2-0.5 Hz.

➔ Thank you for the good comment. As responded to the other reviewer's major comment, we additionally conducted the sensitivity test on the inertial range. When the EDRs with the fixed inertial range (EDR1, EDR2, and EDR3) and those with dynamically selected range (EDR1-opt, EDR2-opt, and EDR3-opt) that minimizes the discrepancy between the theoretical and observed power laws within a given range are computed, it is found that there exist high correlations (r > 0.97 for the B777) and low MAEs (between 0.001-0.003 $m^{2/3}$ $s^{-1}$ for B777) between EDRs using the fixed range and dynamically selected range (Figure A5). In the present

study, the fixed inertial range is considered, regardless of an underestimation in the magnitude of some EDRs (e.g., EDR1U and EDR1V), as it can be more computationally efficient in calculating the EDR. Further investigation, however, may be required using more and longer data in the future. This statement is included and Fig. A5 is included as a new figure (Figure 4) in the revised manuscript. [Page 8, Lines 4-13]

[Figure]

**Figure A5.** Scatter density plots of EDRs using the fixed and dynamical inertial ranges for the B777 data. Pearson correlation and MAE are given in the top-left corner of each panel.

12) Figure 4: The labels on Fig. 4 are confusing. The two models can be -5/3 since the lines are not parallel in the plot. Also, it would be good to add what the actual EDR values are in the figure for the readers benefit. Also, the B737 will have a Nyquist frequency of 0.125 Hz, which is out of the time interval for the -5/3 fit that the authors propose (2 – 5 s).

➔ Figure 4 is modified to Figure 6 of the revised manuscript, as suggested.

13) Page 8, Lines 4-5: Not sure this is reasonable to state without further proof.

➔ The sentence is deleted during the revision process.

14) Page 9, Lines 9-12: These equations are redundant, as they are the exact same as in Eq. 11. Please remove.

➔ The equations are deleted as suggested.

15) Figure 5: I recommend the plot the x-axis (DEVG) in log scale, since that will help evaluate the nature of the fits and PDFs better. Also, it seems only half of the distribution is plotted. The point of the log-normal distribution is that there are instances to the left of the mean of the distribution (lower values), which are symmetric to the larger values (see Sharman and Pearson 2017, their Fig). However, this shows the peak occurs at the lowest DEVG values. It seems like there may be better distributions to use that the log-normal for DEVG, which may potentially lead to a better fit.

➔ The x-axis of Fig. 7 is changed into the log scale in the revised manuscript as suggested. Also, a whole distribution was plotted in Figure 5 of the original manuscript, which is a similar plot to Figure 9 of Kim et al. (2020).

16) Page 9, Line 20: "wide coverage". I assume you mean over time, but it would be good to specify.

➔ Thank you for the comment. The statement is modified in the revised manuscript. [Page 10, Lines 11-12]

17) Figures 6 and 7: Here the x-axis needs to be in log-scale. Many of the distributions do not look log-normal, but maybe the x-log will help. Also, please use the same range of log(EDR) for all panels. Otherwise, it is very difficult to compare them.

➔ The x-axis of Figures 8 and 9 is changed into the log scale in the revised manuscript as suggested. Also, although we respect the reviewer's suggestion, we decided to keep the different range of the EDR for better representation. Instead, we include the notification of different range of the x-axis in the figure caption.

18) Page 15, Lines 12-31: This appears to be out of the scope and context of the current manuscript, since forecasting is not the topic of this work.

➔ Thank you for your comment. As one of other reviewers (Reviewer 2) recommended that it

is worth to discuss, we decide to keep this context in the revised manuscript with modifications to clearly point out the messages in summary. As shown in the case analyses in this study, the characteristics of the EDR observations from different wind components (U, V, and W) are highly depending on the possible sources [Convectively induced turbulence (CIT), clear air turbulence (CAT), and mountain wave turbulence (MWT)] of turbulence in the upper troposphere and lower stratosphere. This can be eventually useful for the situational awareness of cruising aircraft and tactical avoidance for turbulence, and for producing a better climatology of aviation turbulence. These statements are included in the revised manuscript. [Page 16, Lines 5-24]

**References**

Gao, F., and Han, L.: Implementing the Nelder-Mead simplex algorithm with adaptive parameters. Comput. Optim. Appl., 51, 259-277, 2012.

Gill, P. G.: Objective verification of World Area Forecast Centre clear air turbulence forecasts, Meteor. Appl., 21, 3-11, 2014.

Press, W. H., Flannery, B. P., Teukolsky, S. A., and Vetterling, W. T.: Numerical Recipes: The Art of Scientific Computing. 2nd ed. Cambridge University Press, 963 pp., 1992.

Kim, J., Kim, J.-H., and Sharman, R. D.: Characteristics of energy dissipation rate observed from the high-frequency sonic anemometer at Boseong, South Korea, Atmosphere, 12, 837, 2021.

Kim, S.-H., Chun, H.-Y., Kim, J.-H., Sharman, R. D., and Strahan, M.: Retrieval of eddy dissipation rate from derived equivalent vertical gust included in Aircraft Meteorological Data Relay (AMDAR). Atmos. Meas. Tech., 13, 1383-1385, 2020.

---

## Author Comment (AC4)

**[ Responses to the Comment by the Anonymous Referee #3 ]**

>> We deeply appreciate the referee #3 for providing constructive comments. The manuscript is revised following the comments below.

**General comment:**

The paper provides a nice and detailed analysis of energy dissipation rate (EDR) calculated with several approaches using 1 year of data collected by two commercial aircrafts. The paper is well written, the figures generally well made (but please see my comments below), and the analysis is explained in detail. While this reviewer is not an expert in aviation meteorology, I am sure the topic is of great interest for the community. My review focused on the turbulence calculations and boundary layer meteorology aspects of the analysis, and from that point of view I find the paper worth of being accepted after minor revisions.

**Major comments:**

1) Most of the abstract explains the methodology applied to derive EDR, which are not novel or unique to this analysis. I recommend re-shaping the abstract so that more focus is given to the results of this analysis, and especially its novelty aspects compared to previous studies in the same field.

➔ Thank you very much for the good comment. We reconstructed the abstract to highlight results of analyses and novelty aspects of the current study in the revised manuscript. [Page 1, Lines 7-27]

2) Page 6, Line 22: How did you select the "predefined inertial range"? Was this selection valid at both high and low turbulence regimes? At all heights? For both aircrafts? This is critical choice that impacts the EDR calculation, and therefore should be explained in more detail.

➔ Thank you for the good comment. We selected the inertial range that minimizes discrepancy between the theoretical slope and observed one for both types of aircraft, whole altitude ranges (above 15 kft), and both high and low turbulence regimes. This statement is included in the revised manuscript. [Page 6, Lines 23-26]

   During the revision process, we additionally conducted a sensitivity test on the inertial range. Instead of using a fixed inertial range, the inertial range is dynamically selected for each flight data and EDRs are calculated based on current three estimation methods (EDR1, EDR2, and

EDR3) and compared to the EDR in the fixed inertial range. It is noted that a dynamical inertial range is determined by finding the range has the minimum error between the observed power laws and theoretical one (i.e., $s^{2/3}$ and $f^{5/3}$) for a given time segment. EDRs are calculated based on the three EDR estimations using the dynamical inertial range (EDR1-opt, EDR2-opt, and EDR3-opt) and resultant EDRs are compared to EDRs using the predefined (fixed) inertial range (EDR1, EDR2, and EDR3). Pearson correlation (r) and mean absolute error (MAE) between two different EDRs are also computed. It is found that there exist high correlations more than 0.97 and low MAEs 0.001-0.003 $m^{2/3}$ $s^{-1}$ between EDRs using the fixed range and dynamically selected range for Boeing (B) 777 data (Fig. A1). For B737 data, we found r = 0.93 and MAE = 0.002-0.007 $m^{2/3}$ $s^{-1}$ (not shown). In the present study, the fixed inertial range is considered, regardless of an underestimation in the magnitude of some EDRs (e.g., EDR1U and EDR1V), as it can be more computationally efficient in calculating the EDR. Further investigation, however, may be required using more and longer data in the future. This statement is included and Fig. A1 is included as a new figure (Figure 4) in the revised manuscript. [Page 8, Lines 4-13]

[Figure]

**Figure A1.** Scatter density plots of EDRs using the fixed and dynamical inertial ranges for the B777 data. Pearson correlation and MAE are given in the top-left corner of each panel.

**Specific comments:**

1) Page 2, Line 22: Should be "Southwest". Also, these references to commercial airlines probably require a citation.

➔ Thank you for the comment. The phrase is modified, and references are included in the revised manuscript. [Page 2, Lines 23-25]

2) Page 4, Line 5: The following sentence is confusing and somewhat misleading; you should just say that the B737 data were recorded at 0.25 Hz. The same type of misleading information is used many times in the rest of the paper. "The aircraft flight parameters used in the current study were recorded every second (1 Hz) for both the B737 and B777. Because the 1-Hz wind direction and wind speed of the B737 records had the same values within a 4-second time window, the wind direction and wind speed of the B737 records seem to have 1/4-Hz sampling frequency."

➔ We are sorry for if the statement of the original manuscript brought a confusion. The B737 data is also recorded every 1-Hz. The related statement is modified in the revised manuscript to avoid any confusion. [Page 4, Lines 4-5]

3) Page 6, Line 3: Please delete the following sentence as you have already stated this piece of information: "(264,867 and 1,065,855 reports from B737 and B777 recorders, respectively)".

➔ The sentence is deleted as suggested.

4) Eq. 5: Please define what the brackets "<>" mean.

➔ Thank you for pointing out this. The bracket notation is defined in the revised manuscript. [Page 6, Line 22]

5) Page 15, Line 2: missing ")".

➔ Thank you for pointing out this mistake, parenthesis is included in the revised manuscript. [Page 15, Line 21]

6) Page 16, Line 6: "and so on" is not appropriate for a scientific paper, please rephrase.

➔ The phrase is deleted in the revised manuscript.

7) Figures 9, 11, 13: Can you consider using a log scale on the y-axis of panels b, c, and d? Right now, the plots are very hard to read.

➔ Thank you for the suggestion. The y-axis of Figs. 11, 13, 15 (b), (c), and (d) is changed into the log scale in the revised manuscript.

8) Figure 12: It's hard to see the black circle(s), please use a different color.

➔ The color of circles of Fig. 14 is changed in the revised manuscript.

9) Throughout the paper, please consider using a different color scheme. The rainbow color has quite a lot of issues, see for example https://www.climate-lab-book.ac.uk/2014/end-of-the-rainbow/.

➔ Thank you for the good suggestion. We use a different color scheme instead of the rainbow color throughout all figures of the revised manuscript.

---

## Author Comment (AC5)

**[ Responses to the Comment by the Anonymous Referee #2 ]**

>> We deeply appreciate the referee #2 for providing constructive comments. We carefully addressed all comments and tried our best to improve the manuscript based on the referee's suggestions and comments.

EDR estimation in aircraft continues to be an issue for commercial aircraft and the relationship between good observation data and applicable use for safety and integration into planning such as meteorological impacts such as numerical modeling. As the author notes, EDR is the ICAO official observing unit for turbulence in aircraft. This is an important point in all of this work, as it increases the relevancy of this work. One of the biggest hurdles is finding an efficient way to calculate this data for safety applications. Many commercial airlines use various observing methods, not EDR, to communicate turbulence impacts internally. One of these is the DEVG unit of measurement that the author uses as a basis for some of their EDR estimations.

**Suggestions:**

1-1) The first suggestion is clarifying what and where the QAR data is retrieved. Previous papers have used QAR data, but are quick to point out that this data is retrieved post-flight and do not have a real time application.

➔ Thank you for your good suggestion. As the reviewer mentioned, the quick access recorder (QAR) data used in this study is not the real-time data but the retrieved post-flight data. This is clarified in the manuscript. [Page 3, Line 26; Page 14, Lines 28-29]

1-2) Also, the author should explain up front what the advantage to 1-Hz data estimations have over current operational EDR observations done at 10-Hz.

➔ Thank you for pointing out this. The high-frequency EDR data is considered as a truth of atmospheric turbulence measured by aircraft in terms of intensity and location, because it can capture highly transient and intermittent small-scale turbulence hazardous to cruising aircraft. However, when the high-frequency in-situ aircraft measurement is not available, 1-Hz data can be used as an additional source for measuring the EDR. This can contribute to expand more EDR information at the cruising altitudes in the upper troposphere and lower stratosphere (UTLS) in the world. A similar attempt can be made for other lower-frequency QAR and/or other navigational information of commercial aircraft such as Mode-S Enhanced Surveillance

(EHS) and Automatic Dependent Surveillance-Broadcast (ADS-B) in the future. This statement is included in the manuscript. [Page 3, Lines 16-18, 20-24; Page 15, Lines 17-19; Page 16, Lines 23-27]

1-3) Is the cost/loss of higher frequency data to support commercial safety enough to continue to advance this?

➔ Thank you for pointing out this. The high-frequency (e.g., 8 Hz or 10 Hz) aircraft data has been used to estimate EDR, which can capture highly transient and intermittent of small-scale turbulence hazardous to cruising aircraft. For instance, the high frequency National Center for Atmospheric Research (NCAR) EDR algorithm has been developed and implemented in some US-based commercial aircraft, and will be extended to more airliners worldwide in the future (e.g., Sharman et al. 2014; Cornman 2016). In this paper, we focused more on the additional EDR estimations from the currently available source of aircraft data. This statement is included in the manuscript. [Page 3, Lines 16-20, 24-25]

1-4) The results presented would indicate it is, but would need to be tested in a real-time experiment.

➔ We totally agree that a real-time experiment is needed. Currently, the main purpose of this study is trying to find out an additional source of EDR estimations by applying all possible EDR methods to different wind component (U, V, W) of 1-Hz (coarser frequency) post-flight data including some of different cases [Clear air turbulence (CAT), mountain wave turbulence (MWT), and convectively induced turbulence (CIT)] of atmospheric turbulence in cruising altitudes. This test can be extended by applying these EDR estimation methods to the 1-Hz real-time data in future, which can be eventually useful for a better awareness of atmospheric condition for cruising aircraft. Possible candidates for the 1-Hz real-time data are the navigational information of commercial aircraft such as the Mode-S EHS and ADS-B. This statement is included in the manuscript. [Page 3, Lines 21-25; Page 15, Lines 17-19; Page 16, Lines 23-27]

2) In section 2.2, since data sampling is limited, is this an optimal percentage of data used to calibrate? Why were those percentage of data chosen?

➔ Thank you for pointing out this. There is a misunderstanding in this part. To objectively retrieve the intensity of atmospheric turbulence using the aircraft data (especially using vertical

velocity; W), we should find the best relationship between the measured angle of attack ($\bar{\alpha}$) and aircraft pitch angle ($\theta$) for estimating the derived W by the Eq. (4) of the manuscript. Because two parameters (pitch angle and angle of attack) are highly sensitive to the navigation (maneuvering) of aircraft, we tried to extract any data that are not related to the altitude changes by the lower limits of altitude and altitude rate as 15 kft and 10 ft/s, respectively. Using this criterion, we found that most of the flight data [81% of Boeing (B)737 and 94% of B777] were in the cruising mode of the steady flights, which are eventually used to construct the best linear regression between the measured angle of attack ($\bar{\alpha}$) and aircraft pitch angle ($\theta$) for estimating the derived W by the Eq. (4). After this, we eventually use all of the 1-Hz of zonal (U), meridional (V), and derived W data above 15 kft for estimating various EDRs. To avoid any confusion, the statements are modified in the manuscript. [Page 5, Lines 20-26]

3) Section 3.3, the author uses the climatological values from Sharman and Pearson (2017). This dataset has become too outdated as the number of EDR equipped aircraft that could be used for those calculations and perhaps those numbers should be updated.

➔ Thank you for the comment. Considering that the 1-year (2012) period of the QAR data used in this study overlaps the research period (from 2009 to 2014) of the dataset used in Sharman and Pearson (2017), we think that the use of climatological values of Sharman and Pearson (2017) is acceptable. Although in recent days there are some efforts to update $C_1$ and $C_2$ for the low-level turbulence using high-frequency sonic anemometer mounted in the tall towers (e.g., Muñoz-Esparza et al. 2018; Kim et al. 2021a), to the knowledge of the authors at the present, there is no recent update on $C_1$ and $C_2$ for the upper level because it requires a large amount of high-frequency aircraft data for the EDR estimation. This study can be one of these efforts to provide more EDR data that are required to update $C_1$ and $C_2$ based on all available flight information including relatively low-frequency flight data such as 1-Hz post-flight data. The statements related to this are also included in the manuscript. [Page 9, Lines 26-30; Page 10, Lines 1-3]

4) Also, the degree of work down to the python library used, might be a little too much information for this paper.

➔ Thank you for the comment. This sentence is deleted in the manuscript.

5) Onward in 3.4, up to this point, it appears this paper is more focused on comparing estimation

methods than what the benefits of the 1-Hz data represents. Perhaps a title change or additional information on the 1-Hz calculations would suffice.

➔ Thank you for the comment. We include the statements related to the benefits of the 1-Hz data in the manuscript as responded above [Page 3, Lines 20-24; Page 15, Lines 17-19; Page 16, Lines 23-27]. Because we have the comparison of the EDR methods and characteristics of EDR for possible sources, the title and subtitle of section 3.4 were slightly changed in the revised manuscript.

6) Section 4, for the convective case, is there any lightning data that would make this more likely due to convection? This would allow you to change verbiage from likely to certainty.

➔ Thank you for the comment. We tried to get the lightning data from Central Weather Bureau of Taiwan. However, they only archive lightning data for recent 12 hours. Therefore, we calculate the minimum brightness temperature near the location of turbulence encounter using 3-hourly GridSat-B1 dataset that has a spatial resolution of 0.07° (Knapp et al. 2011). It is found that the minimum brightness temperature is the lowest at 1800 UTC which is the closest time with the turbulence encounter (Fig. A1). This implies a rapid increase of cloud top height and corresponds well to the COMS satellite images of Fig. 8 of manuscript that show a single-cell-type convection. Therefore, this study considers this turbulence case as the CIT. This statement is included and relevant statements are modified in the manuscript. [Page 12, Lines 3-6]

[Figure]

[Figure]

**Figure A1.** (a) Location of turbulence encounter (asterisk) and 3-point by 3-point domain centered on the observed location (box), and (b) the minimum brightness temperature calculated in domain of (a) using 3-hourly GridSat-B1 data from 1200 to 2100 UTC 20 September 2012.

7-1) Overall message from the case studies is that the EDR estimations are impacted more or less based on the synoptic or mesoscale regimes that form the base.

➔ Thank you very much for your clarification. Yes, the results in this paper strongly suggest that the observed EDR estimates derived from different wind components such as U, V, and W can show different characteristics depending on the potential sources of atmospheric turbulence in the cruising altitudes (UTLS). This is of interest because it can provide a basic information for the classification of the recent in situ EDR from the aircraft-based observation (ABO) data that are useful for producing a better climatology of upper-level turbulence and turbulence forecast systems (e.g., Sharman et al. 2006; Kim et al. 2011, 2018, 2019a, 2021b; Kim and Chun 2016; Sharman and Pearson 2017). Statements with more discussions related to this are included in the manuscript. [Page 14, Line 23; Page 15, Lines 19-24]

7-2) Is there any work done to validate which ones are more accurate?

➔ As far as we know, there is no work done to compare the accuracy of EDR among the EDR estimation methods in the different synoptic and mesoscale regimes, which needs for further investigation in the future. This statement is included in the revised manuscript [Page 14, Lines 23-25].

7-3) Is the author inferring that different EDR methods should be used for different situations, similar to radar changing scanning modes for different weather? This is a fascinating conclusion and I think worth a lot more discussion.

➔ No, this study does not tell that the different EDR methods show different characteristics of the observed EDR. When we used all available methods to estimate EDR using 1-Hz data, it is found that there is no significant difference depending on the different EDR methods. However, this study emphasized that the different EDR values from various wind components such as U, V, and W show significantly different characteristics in the same EDR method. As shown in the case analyses in this study, the characteristics of the EDR observations from different wind components are highly depending on the sources (CIT, CAT, and MWT) of turbulence in the UTLS. This can be eventually useful for situational awareness of cruising aircraft and tactical avoidance for turbulence, and for a better climatology of turbulence classification. The related statements and discussions are included in the manuscript. [Page 15, Lines 25-34; Page 16, Lines 1-22]

8) Also, the author might suggest more how 1-Hz data could be used for in-situ observing and data transmission.

➔ Thank you for your good suggestion. As already mentioned in the response to the reviewer's question 1-4, the current EDR estimations applied to post-flight QAR data should be tested on the real-time wind retrievals. Possible candidates for the real-time data are the navigational information of commercial aircraft such as the Mode-S EHS and the ADS-B. The statements are included in the manuscript. [Page 3, Lines 21-24; Page 16, Lines 23-27]

**References**

[revised manuscript text omitted]

---

## Author Response (AR2)

**[ Responses to the Comment by the Anonymous Referee #4 ]**

>> We deeply appreciate the referee #4 for providing constructive comments. The manuscript is revised following the comments below.

I congratulate the authors for the revisions and improvements they have made to the manuscript based on the comments brought up by this and the other reviewers, and that have alleviated the great majority of my previous concerns with the original manuscript. However, there still is one critical aspect that needs to be addressed by the authors before I can recommend publication of this article. This is related to my previous major comment #2. I have to mention that I appreciate the work the authors have done to compare to a 'true' reference (i.e., a higher-frequency signal) to validate to the EDR estimates out of 1-Hz QAR data.

**Comments:**

The authors now show 2d-histograms comparing EDR1U, EDR2U, and EDR3U methods to their corresponding estimates using the 20-Hz data, presented in the new Fig. 5. However, these histograms exhibit a clear and systematic underestimation when 1-Hz data is used. Therefore, their claim that: "Based on this result, it can be confirmed that the use of 1-Hz flight data to estimate the EDR is reliable, although the magnitude of the EDR is somewhat underestimated in some cases" [Page 8, lines 26-27], is misleading and not accurate. Eyeballing from the figure, the underestimation is ~0.5 of the 20-Hz estimates. This is a non-negligible difference that can cause significant errors, and that the authors must quantify and include in the manuscript. A simple linear regression should suffice, since the bias seems pretty consistent across the EDR range (i.e., parallel to the 1-to-1 line in log-log space). This is in my view one of the most relevant aspects of the paper (if not the most relevant one), since the authors want to show the validity of 1-Hz data to derive EDR, and therefore need to quantify any biases so these can be taken into account when interpreting EDR estimates from the 1-Hz data. This information needs to be given the importance it deserves and discussed in detail in the corresponding section, as well as included in both the abstract and the conclusions (it is a key point of this work!). Also, I would suggest that the authors do not perform any averaging to obtain the equivalent 1-Hz data from the higher frequency signal, and simply decimate it (i.e., pick every 20th sample), since their averaging could be partially contributing to some of the energy underestimation and subsequent reduced EDR values.

➔ Thank you very much for your good comment. As you suggested, we conducted three additional sensitivity tests on selecting raw 20-Hz data to 1-Hz data (Fig. A1), as follows: 1) Reynolds averaging (approach used in the original manuscript), 2) First pick, 3) Middle pick, and 4) Last pick. Results shown in Fig. A2 are about the energy spectra of the raw 20-Hz (black lines) and the selected 1-Hz wind data (blue lines). The energy spectra of 1-Hz data follow a theoretical slope (-5/3), especially in the overlapped range of frequency (0.1-0.5 s$^{-1}$). However, near the tail part of the energy spectra the 1-Hz wind data by selecting the arbitrary pick (blue lines in Figs. A2b, c, and d) show relatively larger powers than that of averaged 1-Hz wind (blue line in Fig. A2a) and even that of raw 20-Hz wind (black line in Fig. A2 c). This is likely related to the aliasing problem, which is eventually shown as relatively higher values of EDR2 and EDR3 than the averaged and original EDR values (Figs. 3Ab and c). This feature is not found in the structure function-based EDR1 method, which is consistent with previous study (e.g., Muñoz-Esparza et al. 2018), suggesting that the EDR1 can slightly reduce the uncertainty of retrieved EDR using 1-Hz data. For the Reynolds averaging 1-Hz data, the retrieved EDRs are underestimated systematically regardless of the EDR methods (Fig. A3).

Finally, the results shown in Figs. A4-A6 are about the statistical comparisons of the EDRs based on the structure function (EDR1; Fig. A4) and energy spectrum [EDR2 (Fig. A5) and EDR3 (Fig. A6)] using the 20-Hz raw data (x-axis) to those using the 1-Hz wind data (y-axis). It is found that all 1-Hz (both averaging and arbitrary pick) EDRs are highly correlated to the raw 20-Hz EDRs (r > 0.92). However, as already mentioned above, the averaging results of the 1-Hz data show systematic underestimations of EDRs about 8.14% (EDR1), 10.75% (EDR2), and 12.56% (EDR3). In addition, arbitrary pick experiments have systematic underestimations of about 2.17-2.19% in EDR1, and overestimations of about 9.18-9.32% and 10.75-10.91% in EDR2 and EDR3 due to the aliasing problem, respectively.

Given the situation that we do not know whether the 1-Hz QAR data used in this study is the averaged value or arbitrary picked one from the raw flight data, the uncertainties found in this study should be considered when we use the retrieved EDR values from the low frequency (1-Hz) flight data for investigating turbulence encounters. Accordingly, we have modified our original manuscript: in the abstract (Page 1, Lines 14-18), in main section (Page 8, Lines 17, 19, 21-30, 33-34; Page 9, Lines 1-6; Figure 5), in summary section (Page 15, Lines 15-23), and supplementary figure (Fig. S1).

[Figure]

**Figure A1.** Timeseries of streamwise component of wind data (U) obtained from the BMO between 1000 LST and 1030 LST 1 October 2019 at 300 m AGL. Raw 20-Hz BMO (black) data are subsampled to the 1-Hz wind data (a) using Reynolds averaging (red) and by picking every (b) first (blue), (c) middle (orange), and (d) last (green) sample.

[Figure]

**Figure A2.** The energy spectrum of U-wind obtained from raw 20-Hz (black) and subsampled 1-Hz (blue) BMO data at 1025 LST 1 October 2019 at 300 m AGL. The dashed line represents the theoretical slope ($f^{-5/3}$) in the frequency domain. Raw 20-Hz BMO data are subsampled to the 1-Hz wind data (a) using Reynolds averaging and by picking every (b) First, (c) Middle (10th), and (d) Last (20th) sample.

[Figure]

**Figure A3.** As in Fig. A2, but for the (a) EDR1s, (b) EDR2s, and (c) EDR3s.

[Figure]

**Figure A4.** Scatter density plots of the EDR1 calculated using the 20-Hz wind data (x-axis) and the 1-Hz wind data (y-axis) with linear regression line (red dashed line). Raw 20-Hz BMO data are subsampled to the 1-Hz wind data (a) using Reynolds averaging and by arbitrary selections from (b) first pick, (c) middle pick, and (d) last pick within each time window.

[Figure]

**Figure A5.** As in Fig. A4, but for the EDR2.

[Figure]

**Figure A6.** As in Fig. A4, but for the EDR3.

**References**

Muñoz-Esparza, D., Sharman, R. D., Lundquist, J. K.: Turbulence dissipation rate in the atmospheric boundary layer: observations and WRF mesoscale modelling during the XPIA field campaign, Mon. Wea. Rev., 146, 351-371, 2018.